# ON THE FOURIER ANALYSIS IN THE SO(3) SPACE : THE EQUILOPO NETWORK

**Dmitrii Zhemchuzhnikov[1,2]∗ and Sergei Grudinin[1]**

[1] Univ. Grenoble Alpes, CNRS, Grenoble INP, LJK, 38000 Grenoble, France

[2] AIRI

`zhemchuzhnikov@airi.net, sergei.grudinin@univ-grenoble-alpes.fr`

## ABSTRACT

Analyzing volumetric data with rotational invariance or equivariance is currently an active research topic. Existing deep-learning approaches utilize either group convolutional networks limited to discrete rotations or steerable convolutional networks with constrained filter structures. This work proposes a novel equivariant neural network architecture that achieves analytical Equivariance to Local Pattern Orientation on the continuous SO(3) group while allowing unconstrained trainable filters - EquiLoPO Network. Our key innovations are a group convolutional operation leveraging irreducible representations as the Fourier basis and a local activation function in the SO(3) space that provides a well-defined mapping from input to output functions, preserving equivariance. By integrating these operations into a ResNet-style architecture, we propose a model that overcomes the limitations of prior methods. A comprehensive evaluation on diverse 3D medical imaging datasets from MedMNIST3D demonstrates the effectiveness of our approach, which consistently outperforms state of the art. This work suggests the benefits of true rotational equivariance on SO(3) and flexible unconstrained filters enabled by the local activation function, providing a flexible framework for equivariant deep learning on volumetric data with potential applications across domains. Our code is publicly available at `https://gricad-gitlab.univ-grenoble-alpes.fr/GruLab/ILPO/-/tree/main/EquiLoPO`.

## 1 INTRODUCTION

Deep-learning methods have shown remarkable success in analyzing spatial data across various domains. However, in many real-world scenarios, the data can be presented in arbitrary orientations. Therefore, the output of the neural network should be invariant or equivariant to rotations of the input. While data augmentation can partially address this requirement, it leads to increased computational demand, especially for volumetric data, which have three rotation angles.

To tackle this challenge, researchers have developed equivariant neural network architectures that utilize rotationally equivariant operations. We can broadly categorize these methods into two classes: group convolutional networks and steerable convolutional networks. Group convolutional networks achieve rotational equivariance by convolving data in both translational and rotational spaces, but they are typically limited to a discrete set of rotations. On the other hand, steerable convolutional networks employ filters that are analytically equivariant to continuous rotations, but they impose constraints on the filter structures.

In this work, we propose a novel equivariant neural network architecture that combines the strengths of both approaches. Our key contributions are:

1. We present a group convolutional network that achieves analytical equivariance with respect to the continuous rotational space SO(3) by leveraging irreducible representations as the Fourier basis.

---

∗The work for this paper was performed while the author was at Univ. Grenoble Alpes, CNRS, Grenoble INP, LJK; he is currently affiliated with AIRI.

2. Contrary to steerable convolutional networks, our approach does not impose constraints on the filter structures beyond the finite resolution limits.

3. We present a local activation function in the rotational space that provides a well-defined mapping from input function values to output function values, ensuring the preservation of equivariance properties while avoiding the reduction of the architecture to a steerable convolutional network.

By addressing the limitations of existing methods, our approach offers a powerful and flexible framework for analyzing volumetric data in a rotationally equivariant manner without the need for data augmentation or constraints on filter structures.

## 2 RELATED WORK

### 2.1 DEEP LEARNING FOR VOLUMETRIC DATA AND EQUIVARIANCE

In recent years, deep learning has firmly established its place in the analysis of spatial data. Neural networks learn a hierarchy of features by recognizing spatial patterns at various levels. However, in real-world scenarios, multidimensional data are often given in arbitrary orientation and shift, as the coordinate system is not defined. In such cases, it is desirable for the output of the neural network to be independent of the rotation or shift of the input data. While dealing with shifts is straightforward through the use of convolution, which is inherently shift-equivariant, achieving equivariance with respect to rotations requires additional efforts. A primary solution for achieving this effect is augmentation of the training dataset with rotated samples (Krizhevsky et al., 2012). Augmentation leads to an increase not only in the dataset size and consequently training time, but also in the number of network parameters needed to memorize patterns in multiple orientations. In volumetric cases, this increased demand for computational resources can be overwhelming, as there are three angles of rotation in 3D, compared to just one in 2D. For some data types, a canonical coordinate system may be defined (Pagès et al., 2019; Jumper et al., 2021; Igashov et al., 2021; Zhemchuzhnikov et al., 2022). However, in most real-world scenarios, such a coordinate system cannot be identified, and even within a canonical coordinate system, the same local patterns may be encountered in different orientations. These circumstances have led the community to focus on methods that are analytically invariant or equivariant to rotations, utilizing rotationally equivariant operations.

Rotational equivariant methods for regular data can be divided into two groups: group convolution networks and steerable convolutional networks. Our method formally belongs to the first group but without a specific approach to activation in the Fourier space can be reduced to a method from the second group as it is shown in Appendix A. Thus, we will briefly describe below group convolution networks, spherical harmonic networks and activation in the Fourier space.

### 2.2 GROUP CONVOLUTION NETWORKS

The pioneering method from the first class was the Group Equivariant Convolutional Networks (G-CNNs) introduced by Cohen and Welling (2016a), who proposed a general view on convolutions in different group spaces. Many more methods were built up subsequently upon this approach (Worrall and Brostow, 2018; Winkels and Cohen, 2018; Bekkers et al., 2018; Wang et al., 2019; Romero et al., 2020; Dehmamy et al., 2021; Roth and MacDonald, 2021; Knigge et al., 2022; Liu et al., 2022b; Ruhe et al., 2023). Several implementations of Group Equivariant Networks were specifically adapted for regular volumetric data, e.g., CubeNet(Worrall and Brostow, 2018) and 3D G-CNN (Winkels and Cohen, 2018). Methods of this class achieve rotational equivariance by convolving data not only in translational but also in the rotational space. The authors of these methods consider a discrete set of 90-degree rotations, which exhaustively describe the possible positions of a cubic pattern on a regular grid. However, equivariance with respect to this discrete group of rotations does not guarantee equivariance on the continuous group SO(3). Separable SE(3)-equivariant network (Kuipers and Bekkers, 2023) approximates equivariance in SO(3) but lacks analytical equivariance since the authors sample only a finite set of points in the SO(3) space. Analytical rotational equivariance in $3D$ can be achieved by using irreducible representations in the O(2) or the SO(3) group.

## 2.3 Spherical harmonics networks

Cohen and Welling (2016b) introduced steerable networks, a class of methods that use analytically equivariant filters with respect to a particular group of transformations. The first two approaches that applied analytically-equivariant filters to the $3D$ data are the Tensor Field Networks (TFN) (Thomas et al., 2018) and the N-Body Networks (NBNs) (Kondor, 2018). Kondor et al. (2018) presented a similar approach, the Clebsch-Gordan Nets, applied to data on a sphere. Weiler et al. (2018) presented steerable networks for the regular three-dimensional data. The authors deduced a complete equivariant kernel basis where input and output irreducible features have arbitrary degrees. The filter must belong to the subspace of the equivariant kernels. This requirement may limit the expressiveness of the network (Duval et al., 2023; Weiler et al., 2023). To summarize, group convolutional networks do not put constraints on filters but provide equivariance only on a discrete space of rotations. Spherical harmonics networks use irreducible representations to achieve analytical equivariance on the continuous rotational space but imply constraints on the filters. Our method employs irreducible SO(3) representations (Wigner matrices) as the Fourier basis in a group convolutional network. This approach allows us to obtain analytical equivariance and avoid constraints on the filter shape. In a general setup, such a convolution can be reduced to a steerable network, shown in Appendix A, because an SO(3) irrep can be seen as a set of O(2) irreps. However, we introduce a novel activation function in SO(3) that prevents the reduction of the whole architecture to a steerable network. Indeed, our activation is not permutationally invariant with respect to Wigner coefficients of different orders and the same degree and thus cannot be reduced to a steerable network, as we demonstrate in Appendix A.

## 2.4 Activation Operators on Irreducible Representations

Activation functions play a crucial role in neural network architectures as they introduce nonlinear operations. When working with irreducible representations (irreps) in rotationally-equivariant neural networks, it is mandatory to choose activation functions that preserve the equivariance properties. Several approaches, listed below, have been proposed to apply activation functions to irreps. Norm-based activation functions operate on the norm (magnitude) of each irrep, preserving the equivariance properties. The L2 norm of each irrep is computed, and a scalar activation function is applied to the norm. The activated norm is then used to scale the original irrep (Thomas et al., 2018). The same principle was used for the Fourier decomposition of a function in $3D$ by (Zhemchuzhnikov et al., 2022). Gated activation functions introduce learnable parameters to control the activation of each irrep (Weiler et al., 2018). A separate set of learnable weights is used to compute a gating signal, which is then applied to the irrep using element-wise multiplication. Capsule networks use a special type of activation function called the squashing function Sabour et al. (2017). The squashing function scales the magnitude of the output vectors (irreps) to be between 0 and 1 while preserving their direction. Tensor Product (TP) activation is a learnable activation function that operates on the tensor product of irreps (Kondor et al., 2018). It applies a learnable set of weights to the tensor product of the input irreps and then projects the result back onto the original irrep basis.

The listed activation functions introduce non-linearity in equivariant neural networks. Unlike traditional interpretations, we view irreducible representations (irreps) as Fourier coefficients of the SO(3) space, offering a distinct perspective on their role in the network. From this viewpoint, we can classify activation functions as either global or local. Let $f_{\text{in}}$ and $f_{\text{out}}$ represent the input and output functions of an activation operation $\sigma$. We call an activation global if

$$f_{\text{out}} = \sigma(f_{\text{in}}), \tag{1}$$

meaning that the value at any point in $f_{\text{out}}$ depends on the values at all points in $f_{\text{in}}$. Conversely, an activation is local if

$$f_{\text{out}}(x) = \sigma(f_{\text{in}}(x)), \tag{2}$$

for any point $x$. This implies that the value at any point of the output function depends only on the value at the same point in the input function, establishing a direct and unambiguous mapping between input and output values in real space.

To the best of our knowledge, all the previously published activation methods in this domain are *global* (Weiler and Cesa, 2019; Bekkers et al., 2024). This is suboptimal and can lead to noncompact representations and a lack of feature hierarchy learning, which our approach aims to address. By

integrating Wigner matrices within a Fourier function framework in SO(3), our method maps values unambiguously in real rotational space, aiming to approximate the ReLU operator in real space. We further support the significance of locality with computational experiments, showing that models with local mapping significantly outperform those with global mapping in terms of accuracy.

## 3 SE(3) CONVOLUTION

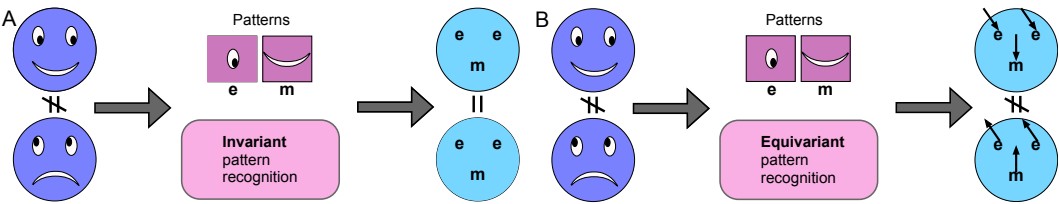

Figure 1: Comparison of invariant and equivariant networks, subplots A and B, respectively. A layer with an invariant pattern recognition cannot distinguish some images, producing identical feature maps for different inputs. In contrast, equivariant pattern recognition allows the discrimination of these images by building feature maps in 6D.

One can think of a six-dimensional (6D) roto-translational convolution that outputs a three-dimensional (3D) feature map using orientational pooling (Zhemchuzhnikov and Grudinin, 2024; Andrearczyk et al., 2020; Kaba et al., 2023). This type of convolution is invariant to orientations of patterns in the input map. However, such a convolution approach is not the most expressive, as it cannot discriminate some input data. Figure 1 schematically demonstrates a toy example where a layer with invariant pattern recognition produces identical feature maps for two smiling and sad faces looking in different directions. A potential solution would be to maintain the output map in 6D: $SE(3)$,

$$h_0(\vec{r}, \mathcal{R}) = \int_{\mathbb{R}^3} d\vec{r}_0 f(\vec{r} + \vec{r}_0) w(\mathcal{R}^{-1}\vec{r}_0), \tag{3}$$

where $f(\vec{r})$ and $w(\vec{r})$ are the input and the filter maps, respectively. This operation is equivariant to both orientations of $f(\vec{r})$ and $w(\vec{r})$ (for more details please see Appendix B). It is worth noticing that the equivariant property in the two cases above holds in different manners. In the first case, both arguments $\vec{r}$ and $\mathcal{R}$ are rotated. In the second case, rotation $\mathcal{R}_0$ is applied only to the second argument. Besides, the expression in Eq. B1 shows how to coordinate rotation of two components of arguments of a 6D map. Introduction of such a 6-dimensional feature map leads to the question how to treat such data. In this paper, we present a novel equivariant convolution in 6D where both input and filter maps are in $SE(3)$ and the set of rotations is continuous,

$$h(\vec{r}, \mathcal{R}) = \int_{\mathrm{SO}(3)} d\mathcal{R}_0 \int_{\mathbb{R}^3} d\vec{r}_0 f(\vec{r} + \vec{r}_0, \mathcal{R}_0) w(\mathcal{R}^{-1}\vec{r}_0, \mathcal{R}^{-1}\mathcal{R}_0). \tag{4}$$

The usage of a continuous space of rotations ensures *analytical* equivariance of the convolution. Below we will show how the irreducible representation helps to perform an integration in the SO(3) rotation space.

### 3.1 INTEGRATION IN THE SO(3) ROTATION SPACE

It is useful to express the rotational part of the convolution in the space of Wigner rotation matrices. Appendix C provides an essential theory on rotational expansions. Let us substitute expansion coefficients from Eq. C8 into the convolution operation $h(\vec{r}, \mathcal{R})$ in Eq. 4 and compute its expansion coefficients:

$$
\begin{aligned}
h_{k_1 k_2}^{l_1}(\vec{r}) &= \frac{2l_1 + 1}{8\pi^2} \int_{\mathrm{SO}(3)} d\mathcal{R} \, h(\vec{r}, \mathcal{R}) D_{k_1 k_2}^{l}(\mathcal{R}) \\
&= \frac{2l_1 + 1}{8\pi^2} \int_{\mathrm{SO}(3)} d\mathcal{R} \, D_{k_1 k_2}^{l_1}(\mathcal{R}) \int_{\mathrm{SO}(3)} d\mathcal{R}_0 \int_{\mathbb{R}^3} d\vec{r}_0 f(\vec{r} + \vec{r}_0, \mathcal{R}_0) w(\mathcal{R}^{-1}\vec{r}_0, \mathcal{R}^{-1}\mathcal{R}_0).
\end{aligned} \tag{5}
$$

Let us now decompose functions $f(\vec{r})$ and $w(\vec{r})$ using Eq. C4, Eq. C2 and Eq. C7,

$$f(\vec{r} + \vec{r}_0, \mathcal{R}_0) = \sum_{l_2=0}^{L_{\text{in}}} \sum_{k_3=-l_2}^{l_2} \sum_{k_4=-l_2}^{l_2} f_{k_3 k_4}^{l_2}(\vec{r} + \vec{r}_0) D_{k_3 k_4}^{l_2}(\mathcal{R}_0),$$

(6)

and

$$w(\mathcal{R}^{-1}\vec{r}_0, \mathcal{R}^{-1}\mathcal{R}_0) = \sum_{l_3=0}^{L_{\text{in}}} \sum_{k_5=-l_3}^{l_3} \sum_{k_6=-l_3}^{l_3} \sum_{k_7=-l_3}^{l_3} \sum_{l_4=0}^{L_{\text{filter}}} \sum_{k_8=-l_4}^{l_4} \sum_{k_9=-l_4}^{l_4}$$
$$w_{k_5 k_7 k_8}^{l_3 l_4}(r) D_{k_5 k_6}^{l_3}(\mathcal{R}_0) D_{k_7 k_6}^{l_3}(\mathcal{R}) D_{k_8 k_9}^{l_4}(\mathcal{R}) Y_{l_4}^{k_9}(\Omega_{\vec{r}_0}),$$

(7)

where $L_{\text{in}}$ and $L_{\text{filter}}$ are the maximum expansion orders of the filter in the rotational and the three-dimensional Euclidean spaces, respectively. Eq. 7 also uses the unitarity property from Eq. C6. Eq. 5 then simplifies to

$$h_{k_1 k_2}^{l_1}(\vec{r}) = \int_{\mathbb{R}^3} d\vec{r}_0 \sum_{l_2=0}^{L_{\text{in}}} \sum_{k_3=-l_2}^{l_2} \sum_{k_4=-l_2}^{l_2} f_{k_3 k_4}^{l_2}(\vec{r} + \vec{r}_0) S_{k_1 k_2 k_3 k_4}^{l_1 l_2}(\vec{r}_0),$$

(8)

where

$$S_{k_1 k_2 k_3 k_4}^{l_1 l_2}(\vec{r}_0) = \frac{8\pi^2}{2l_2 + 1} \sum_{l_4=0}^{L_{\text{filter}}} \left( \sum_{k_5=-l_2}^{l_2} \sum_{k_8=-l_4}^{l_4} \langle l_1 k_2 | l_2 k_5 l_4 k_8 \rangle w_{k_5 k_4 k_8}^{l_2 l_4}(r_0) \right)$$
$$\left( \sum_{k_9=-l_4}^{l_4} \langle l_1 k_1 | l_2 k_3 l_4 k_9 \rangle Y_{l_4}^{k_9}(\Omega_{\vec{r}_0}) \right).$$

(9)

Appendix D presents a more detailed derivation of Eq. 9. Parameters $w_{k_3 k_7 k_8}^{l_2 l_4}(r_0)$ are trainable. The output map expansion coefficients $h_{k_1 k_2}^{l_1}(\vec{r})$ have the maximum degree of $L_{\text{out}}$. We shall also stress that the values of $L_{\text{filter}}$, $L_{\text{in}}$ and $L_{\text{out}}$ must satisfy the triangular inequality,

$$\|L_{\text{in}} - L_{\text{filter}}\| \le L_{\text{out}} \le L_{\text{in}} + L_{\text{filter}}.$$

(10)

The computational complexity of the convolution in Eq. 9 is $O(N^3 L_{\text{filter}}^3 L_{\text{in}}^3 L_{\text{out}}^3 D_{\text{in}} D_{\text{out}})$, where $N$ is the linear size of the input data, and $D_{\text{in}}$ and $D_{\text{out}}$ are numbers of input and output channels. Appendix E proves the roto-translational equivariance with respect to the input data and the rotational equivariance with respect to the filter.

## 4 NON-LINEAR OPERATIONS

In neural networks, linear layers detect patterns and their probabilities, whereas nonlinear layers, specifically activations, selectively amplify these probabilities to form compact representations. We can classify activation functions as either global or local. Let $f_{\text{in}}$ and $f_{\text{out}}$ represent the input and output functions of an activation operation $\sigma$. We call an activation global if $f_{\text{out}} = \sigma(f_{\text{in}})$, meaning that the value at any point in $f_{\text{out}}$ depends on the values at all points in $f_{\text{in}}$. Conversely, an activation is local if $f_{\text{out}}(x) = \sigma(f_{\text{in}}(x))$, for any point $x$. In our work, we aim to propose a local activation function for the Fourier representation of the data in the SO(3) space. Appendix F discusses in more detail why a local activation can be useful. The equivariance of the local activation operator is proven in Appendix H.

### 4.1 LOCAL ACTIVATION IN THE WIGNER SPACE

It has been demonstrated that in NN architectures, on a certain interval, the ReLU function can be approximated by a polynomial of a low degree (Ali et al., 2020; Leshno et al., 1993; Gottemukkula,

2019). Even the quadratic polynomial of degree two showed competitive results in the ResNet architectures (Gottemukkula, 2019). However, as mentioned in Appendix G, the product of two functions defined in the SO(3) space with Wigner coefficients, has a higher resolution than each of the initial functions. Subsequently, to avoid the loss of information, we need to increase the maximum expansion order of the product, given as the Wigner matrix expansion. For a function in SO(3) defined with Wigner coefficients of the maximum degree $L$, the result of applying a polynomial of degree $n$ will have the maximum expansion order of $nL - 1$. Since the number of Wigner expansion coefficients grows as the cube of the expansion order, to approximate the activation function, we chose the activation polynomial of the second degree. This paper studies multiple approximation strategies to this polynomial in the SO(3) space. Generally, the ReLU approximation expression can be written down as

$$f_{\text{act}}(\vec{r}) = D P_2\left(\frac{f(\vec{r})}{D}\right), \tag{11}$$

where $P_2(x) = c_0 + c_1 x + c_2 x^2$ and $D$ is a scaling factor. Below, we discuss several approaches for this approximation in the Wigner space.

In the **adaptive coefficients** approach, we approximate the ReLU operator on a $[-3\sigma(f) + \mu(f), 3\sigma(f) + \mu(f)]$ interval, where $(\mu, \sigma)$ are the mean and the standard deviation of $f(\mathcal{R})$, respectively,

$$\mu = \frac{\int_{\text{SO(3)}} f(\mathcal{R}) d\mathcal{R}}{\int_{\text{SO(3)}} d\mathcal{R}} \equiv f_{00}^0; \;\; \sigma = \sqrt{\frac{\int_{\text{SO(3)}} (f(\mathcal{R}) - \mu)^2 d\mathcal{R}}{\int_{\text{SO(3)}} d\mathcal{R}}} \equiv \sqrt{\sum_{l=1}^{L} \sum_{k_1, k_2 = -l}^{l} \frac{\left(f_{k_1 k_2}^l\right)^2}{2l + 1}}. \tag{12}$$

There are three possible cases to consider. In the special case of $3\sigma + \mu < 0$, we make an assumption that $f(\mathcal{R}) < 0, \forall \mathcal{R}$. Then, we put the polynomial approximation to $y = 0.01x$, instead of $y = 0$ to avoid zero gradients. In the second special case of $-3\sigma + \mu > 0$, we assume $f(\mathcal{R}) > 0, \forall \mathcal{R}$ and set $y = x$. In the general case, the function $f(\mathcal{R})$ ranges both positive and negative values. For simplicity, we first divide the values of the function by $3\sigma$ and then apply a polynomial that approximates ReLU on $[-1 + k, 1 + k]$, where $k = \frac{\mu}{3\sigma}, k \in [-1, 1]$. To find the optimal polynomial coefficients in this case, we formulate the following minimization problem,

$$\min_{c_0(k), c_1(k), c_2(k)} F(c_0, c_1, c_2, k), \tag{13}$$

where

$$F(c_0, c_1, c_2, k) = \int_{-1+k}^{0} dx (c_0 + c_1 x + c_2 x^2)^2 + \int_{0}^{1+k} dx (c_0 + c_1 x + c_2 x^2 - x)^2. \tag{14}$$

Then, we can consistently deduce (see details in Appendix I)

$$c_2 = \frac{15}{32}(k^4 - 2k^2 + 1); \;\; c_1 = \frac{1}{16}(-15k^5 + 26k^3 - 3k + 8); \;\; c_0 = \frac{3}{32}(5k^6 - 9k^4 + 3k^2 + 1). \tag{15}$$

Figure I1 in Appendix I shows the plot of these coefficients as a function of normalized mean $k$ and the error of this approximation, respectively. Finally, to obtain the polynomial activation, we multiply the result by $3\sigma$,

$$f_{\text{act}}(\mathcal{R}) = 3\sigma P_2\left(\frac{f(\mathcal{R})}{3\sigma}\right). \tag{16}$$

In the second strategy with **constant coefficients**, we consider the denominator $D$ equal to the third of the function norm $\|f\|_2$. In practice, this means that the function range lies in the $[-1, 1]$ interval. This is a special case of the interval from the previous approach at $k = 0$. Such a value of $k$ leads to the following polynomial coefficient values,

$$c_2 = \frac{15}{32}; c_1 = \frac{1}{2}; c_0 = \frac{3}{32} \rightarrow P_2 = \frac{3}{32} + \frac{1}{2}x + \frac{15}{32}x^2. \tag{17}$$

We then multiply the result of the polynomial function by $\|f\|_2/3$,

$$f_{\text{act}}(\mathcal{R}) = (\|f\|_2/3) P_2\left(\frac{f(\mathcal{R})}{(\|f\|_2/3)}\right). \tag{18}$$

We have also studied an approach with **trainable polynomial coefficients**. Here, the denominator $D$ is the same as in the previous case, $D = \|f\|_2/3$. However, the three polynomial coefficients $(c_0, c_1, c_2)$ are trainable values.

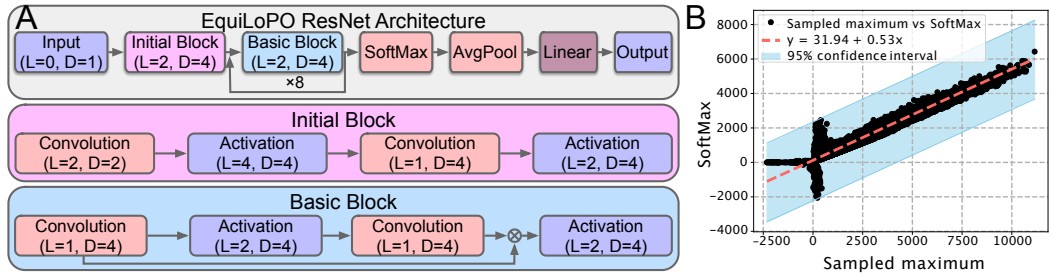

Figure 2: **A** – Schematic representation of the EquiLoPO ResNet-18 architecture, with a sequence of operations in the Initial and Basic blocks. $L$ is the maximum expansion degree of the last operator in the block, and $D$ is the number of the block's features. **B** – Correlation between the Sampled Maximum and its SoftMax approximation of the output functions in the rotational space in the trained model with locally adaptive coefficients on the Vessel dataset.

## 4.2 OTHER NON-LINEAR OPERATIONS

We incorporate additional non-linear operations essential for our framework, including $SO(3)$ pooling, global activation functions, and normalization adapted to rotational spaces. Detailed descriptions are provided in Appendix J.

## 5 RESULTS

Our method is centered on applications involving *regular volumetric data*. Extending this method to irregular data would necessitate significant modifications that are beyond the scope of this paper. Consequently, we evaluated our method using a collection of voxelized 3D image datasets. For this, we used MedMNIST v2, a vast MNIST-like collection of standardized biomedical images (Yang et al., 2023). It is designed to support a variety of tasks, including binary and multi-class classification, and ordinal regression. The dataset encompasses six sets comprising a total of 9,998 3D images. All images are resized to $28 \times 28 \times 28$ voxels, each paired with its respective classification label. We used the train-validation-text split provided by the authors of the dataset (the proportion is $7 : 1 : 2$).

We constructed multiple architectures with various activation methods, each reflecting the layer sequence of ResNet-18 (He et al., 2016). Figure 2A schematically illustrates our EquiLoPo model along with its main components. These include the initial block, the repetitive building block, pooling operators in SO(3) and 3D spaces, and a linear transformation at the end. Appendix K provides details on these blocks. We also specifically adapted the Batch Normalization process for $6D$ ($3D \times SO(3)$) data.

Table 1 lists a detailed comparison of our models' performance against the baseline models: various adaptations of ResNet (He et al., 2016), featuring 2.5D/3D/ACS (Yang et al., 2021) convolutional layers, alongside open-source AutoML solutions such as auto-sklearn (Feurer et al., 2019), and AutoKeras (Jin et al., 2019). We have also added results of the models that were tested on the collection: FPVT (Liu et al., 2022a), Moving Frame Net (Sangalli et al., 2023), Regular SE(3) convolution (Kuipers and Bekkers, 2023) and ILPOResNet50 ((Zhemchuzhnikov and Grudinin, 2024)). They are also discussed in Appendix K.

In MedMNIST, the classes are imbalanced, meaning that to measure the predictive precision of a model, we need other metrics besides accuracy (**ACC**). Consequently, we also consider the AUC-ROC (**AUC**) metric, which provides a more informative assessment of imbalanced datasets. According to the metrics, we can observe that our models either outperform or demonstrate state-of-the-art levels rounded to the third significant digit on nearly all metrics, except for the accuracy on the organ dataset. The organ dataset is largely resolved, with all tested methods achieving a very high level of AUC. Additionally, the dataset might benefit from more channels (more than the four we used) within the model's layers for improved performance. However, one of our goals was to limit the number of parameters in our model.

Among the state-of-the-art methods, only ILPOResNet (Zhemchuzhnikov and Grudinin, 2024) has fewer parameters. The ILPOResNet model is based on the same principles of pattern definition and detection but carries out recognition in an invariant manner while remaining in 3D. In contrast, the presented equivariant model offers a more expressive architecture, as evidenced by its performance on the collection datasets, but requires more parameters.

When considering the various types of activation we tested, three stand out with the best performance: local activation with trainable coefficients, local activation with adaptive coefficients, and global activation with trainable coefficients. Tables L1, L2, and L3 in Appendix L show the results of experiments on architectures with these types of activations but a smaller number of building blocks (1 or 2 instead of 8). Local activation with adaptive coefficients demonstrates the highest robustness since, even in small models, it performs relatively well.

Here, a reasonable question may arise whether the local activation is necessary if the global activation already demonstrates comparable results. We shall notice, however, that global activation, in the current implementation, employs SoftMax in the argument of the multiplier function, which in turn utilizes the ReLU approximation, a form of local activation. In Appendix M, we tested an alternative strategy for global activation, replacing SoftMax with the 2-norm. As evidenced by the results presented in Table M1, this alternative strategy yields significantly inferior performance compared to the original approach with SoftMax.

| Methods | # of prms | Organ AUC | Organ ACC | Nodule AUC | Nodule ACC | Fracture AUC | Fracture ACC | Adrenal AUC | Adrenal ACC | Vessel AUC | Vessel ACC | Synapse AUC | Synapse ACC |
|---|---|---|---|---|---|---|---|---|---|---|---|---|---|
| ResNet-18 (He et al., 2016) + 2.5D(Yang et al., 2021) | 11M | 0.977 | 0.788 | 0.838 | 0.835 | 0.587 | 0.451 | 0.718 | 0.772 | 0.748 | 0.846 | 0.634 | 0.696 |
| ResNet-18 (He et al., 2016)+ 3D(Yang et al., 2021) | 33M | **0.996** | **0.907** | 0.863 | 0.844 | 0.712 | 0.508 | 0.827 | 0.721 | 0.874 | 0.877 | 0.820 | 0.745 |
| ResNet-18 (He et al., 2016)+ ACS(Yang et al., 2021) | 11M | 0.994 | 0.900 | 0.873 | 0.847 | 0.714 | 0.497 | 0.839 | 0.754 | 0.930 | 0.928 | 0.705 | 0.722 |
| ResNet-50 (He et al., 2016)+ 2.5D(Yang et al., 2021) | 15M | 0.974 | 0.769 | 0.835 | 0.848 | 0.552 | 0.397 | 0.732 | 0.763 | 0.751 | 0.877 | 0.669 | 0.735 |
| ResNet-50 (He et al., 2016)+ 3D(Yang et al., 2021) | 44M | 0.994 | 0.883 | 0.875 | 0.847 | 0.725 | 0.494 | 0.828 | 0.745 | 0.907 | 0.918 | 0.851 | 0.795 |
| ResNet-50 (He et al., 2016)+ ACS(Yang et al., 2021) | 15M | 0.994 | 0.889 | 0.886 | 0.841 | 0.750 | 0.517 | 0.828 | 0.758 | 0.912 | 0.858 | 0.719 | 0.709 |
| auto-sklearn* (Feurer et al., 2019) | - | 0.977 | 0.814 | 0.914 | 0.874 | 0.628 | 0.453 | 0.828 | 0.802 | 0.910 | 0.915 | 0.631 | 0.730 |
| AutoKeras* (Jin et al., 2019) | - | 0.979 | 0.804 | 0.844 | 0.834 | 0.642 | 0.458 | 0.804 | 0.705 | 0.773 | 0.894 | 0.538 | 0.724 |
| FPVT* (Liu et al., 2022a) | - | 0.923 | 0.800 | 0.814 | 0.822 | 0.640 | 0.438 | 0.801 | 0.704 | 0.770 | 0.888 | 0.530 | 0.712 |
| SE3MovFrNet * (Sangalli et al., 2023) | - | - | 0.745 | - | 0.871 | - | **0.615** | - | 0.815 | - | 0.953 | - | 0.896 |
| Regular SE(3) convolution (Kuipers and Bekkers, 2023) | 172k | - | 0.698 | - | 0.858 | - | 0.604 | - | 0.832 | - | - | - | 0.869 |
| ILPOResNet-50 | 38k | 0.992 | 0.879 | 0.912 | 0.871 | 0.767 | 0.608 | 0.869 | 0.809 | 0.829 | 0.851 | 0.940 | 0.923 |
| Local trainable activation | 418k | 0.991 | 0.866 | **0.923** | 0.861 | 0.727 | 0.563 | 0.876 | 0.792 | 0.950 | **0.958** | 0.965 | 0.878 |
| Local adaptive activation | 418k | 0.977 | 0.767 | 0.916 | 0.871 | **0.803** | 0.613 | 0.885 | 0.805 | **0.968** | **0.958** | **0.984** | **0.940** |
| Local constant activation | 418k | 0.761 | 0.282 | 0.856 | 0.793 | 0.758 | 0.546 | **0.896** | **0.836** | 0.805 | 0.890 | 0.883 | 0.847 |
| Global trainable activation | 113k | 0.960 | 0.654 | 0.904 | **0.881** | 0.751 | 0.600 | 0.828 | 0.768 | 0.958 | 0.945 | 0.848 | 0.807 |
| Global adaptive activation | 113k | 0.735 | 0.180 | 0.674 | 0.784 | 0.567 | 0.438 | 0.745 | 0.785 | 0.680 | 0.887 | 0.671 | 0.727 |
| Global constant activation | 113k | 0.754 | 0.203 | 0.730 | 0.794 | 0.620 | 0.500 | 0.843 | 0.768 | 0.672 | 0.885 | 0.584 | 0.287 |
| Local trainable activation, const. resolution[1] | 548k | 0.944 | 0.598 | 0.739 | 0.794 | 0.624 | 0.379 | 0.507 | 0.768 | 0.709 | 0.885 | 0.620 | 0.345 |

Table 1: Comparison of different methods on MedMNIST's 3D datasets. (*) For these methods, the number of parameters is unknown. ([1]) Here, we implemented the ResNet-50 architecture. In the other designs, we used the ResNet-18 versions.

## 5.1 ACTIVATION ANALYSIS

We previously highlighted the necessity of ensuring that the activation coefficients of the output possess a maximum degree higher than that of the input to prevent information loss when applying the polynomial, specifically, $L_{\text{out}} = 2L_{\text{in}}$. However, investigating the scenario where high output degrees are discarded, aligning the maximum output degree with the maximum input degree ($L_{\text{out}} = L_{\text{in}}$) merits consideration. Figure 3 illustrates the impact of the increased resolution on the

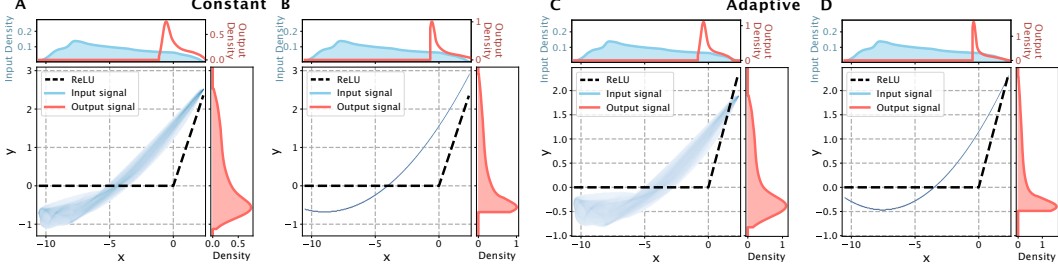

Figure 3: Effect of the ReLU approximation with constant and adaptive polynomial coefficients in the SO(3) space for two resolution strategies: **A,C** – low-resolution setup, $L_{\text{out}} = L_{\text{in}}$; **B,D** – high-resolution setup, $L_{\text{out}} = 2L_{\text{in}}$. Plots **A** and **B** show results with constant activation coefficients. Plots **C** and **D** show results with adaptive activation coefficients. The $x$ and $y$ axes represent the input and output functions, respectively. The histograms above and to the right of the main plot display the distribution of these functions' values. The top histogram plots also compare the input distribution (blue) with the output distribution (red). Opacity of the main plots indicate the density distributions, overlaid with the ReLU functions.

approximation accuracy of the SO(3) ReLU function. For this plot, we analyzed one of the rotational distributions from the last convolution of the trained model with local adaptive coefficients on the Vessel dataset reported in Table 1 ($L_{\text{in}} = 2$) to observe how the activation operator, with constant (subplots **A-B**) and adaptive (subplots **C-D**) polynomial coefficients, operates under two conditions: $L_{\text{out}} = L_{\text{in}}$ and $L_{\text{out}} = 2L_{\text{in}}$. We sampled $10^6$ points in SO(3) and computed the values of both input and output functions at these points, presenting the results on a density plot.

Subplots **A** and **C** offer illustrations for the scenario with a reduced resolution ($L_{\text{out}} = L_{\text{in}}$). Here, we note an inherent ambiguity between the values of the input and output signals. Indeed, each input function value maps to multiple output values. Such an inadequate mapping is also evidenced by the absence of a sharp negative value cutoff. Subplots **B** and **D** showcase the polynomial approximation with an enhanced output resolution. This setup eliminates the ambiguity in mapping in the real space, as each input function value uniquely corresponds to a single output function value. Moreover, this operator exhibits nonlinearity and closely approximates the ReLU function, depicted by a black dashed line. Consequently, negative input values transform into near-zero output values, a phenomenon clearly visible in the right-hand histogram, which features a prominent peak near $y = 0$ and a sharp cutoff of negative values.

The extent to which increasing the resolution enhances the performance of the architecture is evident in Table 1. We conducted additional tests using a single architecture with a reduced resolution. This architecture employs local activation functions with trainable polynomial coefficients. As we can see from the table, the model with reduced resolution consistently underperforms across all datasets compared to the full-resolution model that also utilizes locally trainable coefficients.

In the case when the input function's mean is close to zero, the adaptive coefficients are almost identical to the constant ones and the two strategies have very similar effect as suggested by Eq. 17. In practice, zero-mean functions are frequent but there are also cases when the mean is shifted. Figure 3 demonstrates how the activation acts on a function with the mean significantly shifted to the negative values.

Subplots **A** and **B** demonstrate distribution of the input function defined in SO(3). The center of the distribution is clearly shifted to the left. In this case, the polynomial with adaptive coefficients (subplot **D**) approximates the ReLU function more precisely than the polynomial with constant coefficients (subplot **B**) because the approximation is closer to the ReLU function. Indeed, according to Eq. 14, the adaptive coefficients strategy gives twofold less approximation error than the constant coefficients one.

Table 1 presents the performance metrics for both strategies across the dataset collection. The choice between local and global activation significantly influences the performance. Therefore, our comparative analysis focuses more on contrasting local activations with adaptive versus constant polynomial coefficients. The strategy employing adaptive coefficients outperforms the one with constant coef-

ficients on all the datasets, except for **Adrenal**. This outcome is expected, as adaptive coefficients typically provide a superior approximation of the activation function.

We evaluated global and local activation layers across three distinct strategies for the coefficients of the approximating activation polynomial. Table 1 indicates that for the trainable coefficients, the performance of global activation is comparable to that of the local activation. However, for both constant and adaptive coefficients, the local activation demonstrates superior results across all datasets.

### 5.2 ANALYSIS OF THE SOFTMAX POOLING IN SO(3)

To examine the behavior of the SoftMax operation, we analyzed the outputs from the last convolution layer of the trained model with locally adaptive coefficients on the Vessel dataset reported in Table 1. We extracted the SO(3) function coefficients for all the maps, voxels, and channels, and then computed the SoftMax for these functions using Eq. J2. We sampled $10^6$ points in the SO(3) space and reconstructed the functions at these points. Figure 2B demonstrates the relationship between SoftMax and the sampled maximum. It is evident that there is high correlation between the two functions in the positive region of the sampled maximum.

Negative values in the sampled maximum lead to SoftMax values of zero. Near zero, uncertainty increases along with a wide range of SoftMax values. This phenomenon appears due to the polynomial approximation error. Specifically, the accuracy of the SoftMax approximation hinges on the condition that the mean of the output of the ReLU operation approximation must be greater or equal to zero. Additionally, the mean of the SO(3) distribution, or its $L_1$-norm, should be substantially smaller than the $L_2$-norm. While this condition holds in most instances, there are exceptions where it does not. These cases are clearly visible in Figure 2B as those providing high uncertainty in the SoftMax estimation near zero sampled maximum.

## 6 CONCLUSION

This work introduces a novel equivariant neural network architecture that achieves analytical rotational equivariance on the continuous SO(3) group while retaining the flexibility of unconstrained trainable filters. Our key innovations are a group convolutional operation that leverages irreducible representations as the Fourier basis and a local activation function in the SO(3) space that provides a well-defined mapping from input to output function values in the real space.

**Key Takeaways:**

- **Enhanced Performance through Rotational Equivariance:** By incorporating rotational equivariance, our models consistently outperformed or matched state-of-the-art methods across various datasets in the MedMNIST collection. This underscores the importance of respecting rotational symmetries in volumetric data.
- **Importance of Activation Function Design:** The use of higher-resolution local activation functions with adaptive coefficients significantly improved the network's ability to learn complex patterns, leading to better performance. This highlights the need for carefully designed activation functions in equivariant architectures.
- **Effectiveness of SO(3) Pooling Operations:** The SoftMax pooling operation in SO(3) proved effective in summarizing rotational features, which is essential for global and local prediction tasks.

Our findings emphasize the crucial role of rotation-equivariant operations and appropriate activation functions in deep learning models dealing with 3D data. Future work may explore extending these concepts to irregular data structures or integrating them with other forms of equivariance.

### ACKNOWLEDGEMENTS

This work is partially funded by MIAI@Grenoble Alpes (ANR-19-P3IA-0003).

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

APPENDIX

## A  COMPARISON WITH STEERABLE NETWORKS AND IMPORTANCE OF ACTIVATION

As we have indicated above, the EquiLoPo convolution is defined as

$$h_{k_1 k_2}^{l_1 d_{\text{out}}}(\vec{r}) = \int_{\mathbb{R}^3} d\vec{r_0} \sum_{l_2=0}^{L_{\text{in}}} \sum_{k_3=-l_2}^{l_2} \sum_{k_4=-l_2}^{l_2} f_{k_3 k_4}^{l_2 d_{\text{in}}}(\vec{r}+\vec{r_0})[S_{\text{EquiLoPo}}]_{k_1 k_2 k_3 k_4}^{l_1 l_2 d_{\text{in}} d_{\text{out}}}(\vec{r_0}), \qquad (A1)$$

where

$$[S_{\text{EquiLoPo}}]_{k_1 k_2 k_3 k_4}^{l_1 l_2 d_{\text{in}} d_{\text{out}}}(\vec{r_0}) = \frac{8\pi^2}{2l_2+1} \sum_{l_4=0}^{L_{\text{filter}}} [p_{\text{EquiLoPo}}]_{k_1 k_3}^{l_1 l_2 l_4 d_{\text{in}} d_{\text{out}}}(r_0) \left( \sum_{k_9=-l_4}^{l_4} \langle l_1 k_2 | l_2 k_4 l_4 k_9 \rangle Y_{l_4}^{k_9}(\Omega_{\vec{r_0}}) \right), \qquad (A2)$$

and

$$[p_{\text{EquiLoPo}}]_{k_1 k_3}^{l_1 l_2 l_4 d_{\text{in}} d_{\text{out}}}(r_0) = \sum_{k_7=-l_2}^{l_2} \sum_{k_8=-l_4}^{l_4} \langle l_1 k_1 | l_2 k_7 l_4 k_8 \rangle [w_{\text{EquiLoPo}}]_{k_3 k_7 k_8}^{l_2 l_4 d_{\text{in}} d_{\text{out}}}(r_0). \qquad (A3)$$

In our implementation, we set coefficients $[w_{\text{EquiLoPo}}]_{k_3 k_7 k_8}^{l_2 l_4 d_{\text{in}} d_{\text{out}}}(r_0)$ to be trainable but it is equivalent to the case when $[p_{\text{EquiLoPo}}]_{k_1 k_3}^{l_1 l_2 l_4 d_{\text{in}} d_{\text{out}}}(r_0)$ is trainable because

$$\sum_{l_1=|l_2-l_4|}^{l_2+l_4} \sum_{k_1=-l_1}^{l_1} \langle l_1 k_1 | l_2 k_7 l_4 k_8 \rangle [p_{\text{EquiLoPo}}]_{k_1 k_3}^{l_1 l_2 l_4 d_{\text{in}} d_{\text{out}}}(r_0) = [w_{\text{EquiLoPo}}]_{k_3 k_7 k_8}^{l_2 l_4 d_{\text{in}} d_{\text{out}}}(r_0). \qquad (A4)$$

Steerable Networks in the 3-dimensional case (Weiler et al., 2018) employ the following convolution operator:

$$h_{k_2}^{l_1 d_{\text{out}}}(\vec{r}) = \int_{\mathbb{R}^3} d\vec{r_0} \sum_{l_2=0}^{L_{\text{in}}} \sum_{k_4=-l_2}^{l_2} f_{k_4}^{l_2 d_{\text{in}}}(\vec{r}+\vec{r_0})[S_{\text{Steerable}}]_{k_2 k_4}^{l_1 l_2 d_{\text{in}} d_{\text{out}}}(\vec{r_0}), \qquad (A5)$$

where

$$[S_{\text{Steerable}}]_{k_2 k_4}^{l_1 l_2 d_{\text{in}} d_{\text{out}}}(\vec{r_0}) = \sum_{l_4=0}^{L_{\text{filter}}} [p_{\text{Steerable}}]^{l_1 l_2 l_4 d_{\text{in}} d_{\text{out}}}(r_0) \left( \sum_{k_9=-l_4}^{l_4} \langle l_1 k_2 | l_2 k_4 l_4 k_9 \rangle Y_{l_4}^{k_9}(\Omega_{\vec{r_0}}) \right), \quad (A6)$$

and coefficients $[p_{\text{Steerable}}]^{l_1 l_2 l_4 d_{\text{in}} d_{\text{out}}}(r_0)$ are trainable.

If we consider convolution in isolation from other operations in a network, then our convolution operator is equivalent to the Steerable network convolution where the number of input and output degree features are $D_{\text{in}}^{l_1} = D_{\text{in}}(2l_1+1)$ and $D_{\text{out}}^{l_2} = D_{\text{out}}(2l_2+1)$ respectively:

$$[p_{\text{Steerable}}]^{l_1 l_2 l_4 (d_{\text{in}}(2l_1+1)+k_1)(d_{\text{out}}(2l_2+1)+k_1)}(r_0) = [p_{\text{EquiLoPo}}]_{k_1 k_3}^{l_1 l_2 l_4 d_{\text{in}} d_{\text{out}}}(r_0). \qquad (A7)$$

The difference between our method and Steerable Networks lies in the interpretation of the coefficients, namely we treat them as expansion coefficients in SO(3). This interpretation results in a special approach to activation. We propose such operation on the expansion coefficients that would correspond to the ReLU operator in the rotational space.

## B  EQUIVARIANCE OF THE CONVOLUTION WITH ROTATED FILTER IN 3D

Below we demonstrate equivariance of the 6D convolution to the orientations of the input $f(\vec{r})$ map and the filter $w(\vec{r})$ map. If $f(\vec{r}) \to f(\mathcal{R}_0^{-1}\vec{r})$, then the convolution output

$$h_0(\vec{r}, \mathcal{R}) \to \int_{\mathbb{R}^3} f(\mathcal{R}_0^{-1}(\vec{r}+\vec{r_0}))w(\mathcal{R}^{-1}\vec{r_0})d\vec{r_0}$$

$$= \int_{\mathbb{R}^3} f(\mathcal{R}_0^{-1}\vec{r}+\vec{r_0})w(\mathcal{R}^{-1}\mathcal{R}_0\vec{r_0})d\vec{r_0} = h_0(\mathcal{R}_0^{-1}\vec{r}, \mathcal{R}_0^{-1}\mathcal{R}). \qquad (B1)$$

Consequently, if $w(\vec{r}) \to w(\mathcal{R}_0^{-1}\vec{r})$, then the convolution output

$$h_0(\vec{r}, \mathcal{R}) \to \int_{\mathbb{R}^3} f(\vec{r} + \vec{r}_0)w(\mathcal{R}^{-1}\mathcal{R}_0^{-1}\vec{r}_0)d\vec{r}_0 = h_0(\vec{r}, \mathcal{R}_0\mathcal{R}). \tag{B2}$$

## C  THEORETICAL FRAMEWORK ON SPHERICAL HARMONICS, WIGNER, AND CLEBSCH-GORDAN COEFFICIENTS

This section provides an overview of spherical harmonics, Wigner coefficients, and Clebsch-Gordan coefficients, essential mathematical tools in the field of quantum mechanics for the analysis of angular momentum.

### C.1  SPHERICAL HARMONICS

Real spherical harmonics $Y_l^m(\Omega) : S^2 \to \mathbb{R}$ are a set of orthogonal functions defined on the surface of a sphere, denoted as $S^2$. They are useful in expanding functions defined over the sphere and appear extensively in the solution of partial differential equations in spherical coordinates. The orthogonality of spherical harmonics is expressed as

$$\int_{S^2} d\Omega Y_l^k(\Omega)Y_{l'}^{k'}(\Omega) = \delta_{ll'}\delta_{kk'}, \tag{C1}$$

where $\delta$ stand for the Kronecker delta, signifying that spherical harmonics are orthogonal with respect to both the degree $l$ and the order $m$. Any square-integrable function $f(\Omega) : S^2 \to \mathbb{R}$ can be decomposed into real spherical harmonics as

$$f(\Omega) = \sum_{l=0}^{\infty} \sum_{k=-l}^{l} f_l^k Y_l^k(\Omega), \tag{C2}$$

where

$$f_l^k = \int f(\Omega)Y_l^k(\Omega)d\Omega \tag{C3}$$

are the spherical harmonic expansion coefficients. In practice, we use a fixed maximum expansion order $L$ defined by the resolution of input data.

### C.2  WIGNER COEFFICIENTS

Wigner matrices, denoted as $D_{mm'}^l(\mathcal{R})$, form the irreducible representations of SO(3), the group of rotations in three-dimensional Euclidean space. These matrices are rotation operations for spherical harmonics,

$$Y_l^{k_1}(\mathcal{R}\Omega) = \sum_{k_2=-l}^{l} D_{k_1 k_2}^l(\mathcal{R})Y_l^{k_2}(\Omega), \tag{C4}$$

highlighting the transformation properties of spherical harmonics under rotation. The orthogonality of Wigner coefficients is held due to the following property,

$$\int_{SO(3)} d\mathcal{R} D_{k_1 k_2}^l(\mathcal{R})D_{k_1' k_2'}^{l'}(\mathcal{R}) = \frac{8\pi^2}{2l+1}\delta_{ll'}\delta_{k_1 k_1'}\delta_{k_2 k_2'}. \tag{C5}$$

Another property of a Wigner matrix is its unitarity,

$$D_{k_1 k_2}^l(\mathcal{R}^{-1}) = D_{k_2 k_1}^l(\mathcal{R}). \tag{C6}$$

Any square-integrable function $f(\mathcal{R}) : SO(3) \to \mathbb{R}$ can be decomposed into Wigner matrices as

$$f(\mathcal{R}) = \sum_{l=0}^{\infty} \sum_{k_1=-l}^{l} \sum_{k_2=-l}^{l} f_{k_1 k_2}^l D_{k_1 k_2}^l(\mathcal{R}), \tag{C7}$$

where

$$f_{k_1 k_2}^l = \frac{2l+1}{8\pi^2} \int f(\mathcal{R})D_{k_1 k_2}^l(\mathcal{R})d\mathcal{R} \tag{C8}$$

are Wigner matrix decomposition coefficients. Again, for practical considerations in Eq. C7 we will use a fixed maximum expansion order $L$.

### C.3 CLEBSCH-GORDAN COEFFICIENTS

The Clebsch-Gordan coefficients facilitate the coupling of two angular momenta in quantum mechanics, leading to composite states with well-defined total angular momentum. The interconnection between Clebsch-Gordan coefficients, spherical harmonics, and Wigner matrices is illustrated through the integration of products of spherical harmonics and the transformation properties under rotation:

$$\int_{S^2} d\Omega Y_L^K(\Omega) Y_l^k(\Omega) Y_{l'}^{k'}(\Omega) = \sqrt{\frac{(2l+1)(2l'+1)}{4\pi(2L+1)}} \langle l0l'0|L0\rangle \langle lkl'k'|LK\rangle \tag{C9}$$

$$\int_{SO(3)} d\mathcal{R} D_{k_1 k_2}^l(\mathcal{R}) D_{k_1' k_2'}^{l'}(\mathcal{R}) D_{K_1 K_2}^L(\mathcal{R}) = \frac{8\pi^2}{2L+1} \langle LK_1|lk_1 l'k_1'\rangle \langle LK_2|lk_2 l'k_2'\rangle. \tag{C10}$$

## D FILTER EXPRESSION

In this section, we reveal how the expression in Eq. 9 is obtained:

$$S_{k_1 k_2 k_3 k_4}^{l_1 l_2}(\vec{r}_0) = \frac{2l_1+1}{8\pi^2} \sum_{l_3=0}^{L_{\text{in}}} \sum_{k_5=-l_3}^{l_3} \sum_{k_6=-l_3}^{l_3} \sum_{k_7=-l_3}^{l_3} \sum_{l_4=0}^{L_{\text{filter}}} \sum_{k_8=-l_4}^{l_4} \sum_{k_9=-l_4}^{l_4} \left( \int_{SO(3)} d\mathcal{R}_0 D_{k_3 k_4}^{l_2}(\mathcal{R}_0) D_{k_6 k_7}^{l_3}(\mathcal{R}_0) \right)$$

$$\left( \int_{SO(3)} d\mathcal{R} D_{k_1 k_2}^{l_1}(\mathcal{R}) D_{k_6 k_5}^{l_3}(\mathcal{R}) D_{k_9 k_8}^{l_4}(\mathcal{R}) \right) w_{k_5 k_7 k_8}^{l_3 l_4}(r) Y_{l_4}^{k_9}(\Omega_{\vec{r}}) = \frac{2l_1+1}{8\pi^2} \frac{8\pi^2}{2l_2+1} \frac{8\pi^2}{2l_1+1}$$

$$\sum_{l_3=0}^{L_{\text{in}}} \sum_{k_5=-l_3}^{l_3} \sum_{k_6=-l_3}^{l_3} \sum_{k_7=-l_3}^{l_3} \sum_{l_4=0}^{L_{\text{filter}}} \sum_{k_8=-l_4}^{l_4} \sum_{k_9=-l_4}^{l_4} \delta_{l_2 l_3} \delta_{k_3 k_6} \delta_{k_4 k_7} \langle l_1 k_1|l_3 k_6 l_4 k_9\rangle \langle l_1 k_2|l_3 k_5 l_4 k_8\rangle w_{k_5 k_7 k_8}^{l_3 l_4}(r) Y_{l_4}^{k_9}(\Omega_{\vec{r}})$$

$$= \frac{8\pi^2}{2l_2+1} \sum_{l_4=0}^{L_{\text{filter}}} \left( \sum_{k_5=-l_2}^{l_2} \sum_{k_8=-l_4}^{l_4} \langle l_1 k_2|l_2 k_5 l_4 k_8\rangle w_{k_5 k_4 k_8}^{l_2 l_4}(r_0) \right) \left( \sum_{k_9=-l_4}^{l_4} \langle l_1 k_1|l_2 k_3 l_4 k_9\rangle Y_{l_4}^{k_9}(\Omega_{\vec{r}_0}) \right). \tag{D1}$$

## E EQUIVARIANCE OF THE SE(3) CONVOLUTION

Here we prove the roto-translational equivariance of the convolution operator to the input data and the rotational equivariance to the filter.

### E.1 EQUIVARIANCE WITH RESPECT TO THE INPUT FUNCTION

Let $f(\vec{r}, \mathcal{R})$ be rotated by $\mathcal{R}_1$. Then consider the output of the convolution in Eq. 4:

$$\int_{SO(3)} d\mathcal{R}_0 \int_{\mathbb{R}^3} d\vec{r}_0 f(\mathcal{R}_1^{-1}\vec{r} + \mathcal{R}_1^{-1}\vec{r}_0, \mathcal{R}_1^{-1}\mathcal{R}_0) w(\mathcal{R}^{-1}\vec{r}_0, \mathcal{R}^{-1}\mathcal{R}_0)$$

$$= \int_{SO(3)} d\mathcal{R}_0 \int_{\mathbb{R}^3} d\vec{r}_0 f(\mathcal{R}_1^{-1}\vec{r} + \vec{r}_0, \mathcal{R}_1^{-1}\mathcal{R}_0) w(\mathcal{R}^{-1}\mathcal{R}_1\vec{r}_0, \mathcal{R}^{-1}\mathcal{R}_0)$$

$$= \int_{SO(3)} d\mathcal{R}_0 \int_{\mathbb{R}^3} d\vec{r}_0 f(\mathcal{R}_1^{-1}\vec{r} + \vec{r}_0, \mathcal{R}_0) w(\mathcal{R}^{-1}\mathcal{R}_1\vec{r}_0, \mathcal{R}^{-1}\mathcal{R}_1\mathcal{R}_0) = h(\mathcal{R}_1^{-1}\vec{r}, \mathcal{R}_1^{-1}\mathcal{R}). \tag{E1}$$

### E.2 EQUIVARIANCE WITH RESPECT TO THE FILTER FUNCTION

Let $w(\vec{r}, \mathcal{R})$ be rotated by $\mathcal{R}_1$. Then consider the output of the convolution in Eq. 4:

$$\int_{SO(3)} d\mathcal{R}_0 \int_{\mathbb{R}^3} d\vec{r}_0 f(\vec{r} + \vec{r}_0, \mathcal{R}_0) w(\mathcal{R}^{-1}\mathcal{R}_1^{-1}\vec{r}_0, \mathcal{R}^{-1}\mathcal{R}_1^{-1}\mathcal{R}_0) = h(\vec{r}, \mathcal{R}_1\mathcal{R}). \tag{E2}$$

## F  The need for local activation function

In neural networks, linear operations typically alternate with nonlinear ones; the latter are often referred to as activations. Nonlinear operations are necessary to approximate complex dependencies in the data. However, from the perspective of learning hierarchies of patterns, the two types of operations have different purposes. Indeed, linear layers are tasked with detecting patterns and fetching a quantity proportional to the probability of finding a certain pattern. Nonlinear layers, in turn, penalize low pattern probabilities, allowing only high ones to pass and often serve to learn compact representations. An incorrectly chosen activation can lead to noise accumulation when passing through multiple layers and also suboptimal latent representations. A classical activation solution is ReLU and its variants, which, despite some drawbacks, are the most intuitively understandable activation functions. The vanilla ReLU operator passes only positive values of its input. In convolutional and other networks dealing with spatial data, the critical property that differentiates activation operations from the rest of the network is their *locality*, i.e., the activation is applied to different points in the Euclidean space independently.

In our case, as we have already mentioned, the output of the linear convolution is six-dimensional: $\mathbb{R}^3 \times SO(3)$. Since the $\mathbb{R}^3$ dimension discretizes into voxels, we apply the activation to each voxel separately. The most interesting question, however, is how to design the activation function for the $SO(3)$ data dimension. We define the distribution of values in $SO(3)$ in our architecture with Wigner matrix decomposition coefficients. Each coefficient, similarly to the Fourier series, represents the global properties of the entire $SO(3)$ distribution rather than an individual point in $SO(3)$. However, we aim to design a *local* activation for $SO(3)$ in the space of Wigner coefficients.

A previously used approach (Zhemchuzhnikov and Grudinin, 2024; Cohen et al., 2018) consists in sampling the $SO(3)$ space and requires a transformation from the Wigner representation into the rotation space, an activation in the $SO(3)$ space, and an inverse transformation into the Wigner space. However, this approach loses the *analytical SO(3)* equivariance. It may not be a problem since sampling a sufficiently large number of points in the rotation $SO(3)$ space can guarantee effective equivariance. However, this strategy inevitably leads to an increased computational complexity. Conversely, the proposed Wigner coefficient representation, allows using few values to define a function in $SO(3)$ and retaining analytical equivariance at the same time. There are no analytical expressions for the decomposition coefficients of $\text{ReLU}(f(\mathcal{R}))$ given the spherical harmonics coefficients of $f(\mathcal{R})$. However, we will use the analytical expression of the coefficients of a product of two functions, given in Eq. H5 in Appendix G. It enables us to find an analytical expression for the spherical harmonics coefficients for a polynomial applied to a function in $SO(3)$.

## G  Product of two functions in $SO(3)$

The Wigner matrix decomposition coefficients of a product of two functions defined in $SO(3)$ can be expressed through coefficients of two multiplier functions.

Let functions $f_1(\mathcal{R})$ and $f_2(\mathcal{R})$ have coefficients $[f_1]^{l_1}_{k_1 k_2}$ and $[f_1]^{l_1}_{k_1 k_2}$ and $f_{\text{prod}}(\mathcal{R})$ be the product of $f_1(\mathcal{R})$ and $f_2(\mathcal{R})$:

$$f_{\text{prod}}(\mathcal{R}) = f_1(\mathcal{R})f_2(\mathcal{R}). \tag{G1}$$

Then

$$[f_{\text{prod}}]^{l_3}_{k_5 k_6} = \sum_{l_1=0}^{L_1} \sum_{l_2=0}^{L_2} \sum_{k_1=-l_1}^{l_1} \sum_{k_2=-l_1}^{l_1} \sum_{k_3=-l_2}^{l_2} \sum_{k_4=-l_2}^{l_2} \langle l_3 k_5 | l_1 k_1 l_2 k_3 \rangle \langle l_3 k_6 | l_1 k_2 l_2 k_4 \rangle [f_1]^{l_1}_{k_1 k_2} [f_2]^{l_2}_{k_3 k_4} \tag{G2}$$

where $[f_{\text{prod}}]^{l_3}_{k_5 k_6}$ are coefficients of $f_{\text{prod}}(\mathcal{R})$, $L_1$ and $L_2$ are maximal degrees of $f_1$ and $f_2$ respectively. In order to avoid loss of information the maximal degree of $f_{\text{prod}}$ is $L_3 = L_1 + L_2$. Thus, the product operation requires increase of resolution of data.

## H EQUIVARIANCE OF THE LOCAL ACTIVATION OPERATOR

We define our local activation operator by

$$f_{\text{act}}(\vec{r}) = D\, P_2\Big(\frac{f(\vec{r})}{D}\Big), \tag{H1}$$

with

$$P_2(x) = c_0 + c_1 x + c_2 x^2. \tag{H2}$$

Thus,

$$f_{\text{act}}(\vec{r}) = c_0 D + c_1 f(\vec{r}) + \frac{c_2}{D} f^2(\vec{r}). \tag{H3}$$

The equivariance of the constant term $c_0 D$ and the linear term $c_1 f(\vec{r})$ is immediate. The nontrivial part is to show that the quadratic term $f^2(\vec{r})$ is also equivariant.

Let us express the function $f(\vec{r})$ with its Wigner expansion,

$$f(\vec{r}) = \sum_{l=0}^{L} \sum_{k_1,k_2=-l}^{l} [f]^l_{k_1 k_2}\, D^l_{k_1 k_2}(\mathcal{R}), \tag{H4}$$

so that its square (i.e., product with itself) is expressed via

$$[f^2]^{l_3}_{k_5 k_6} = \sum_{l_1=0}^{L_1} \sum_{l_2=0}^{L_2} \sum_{k_1=-l_1}^{l_1} \sum_{k_2=-l_1}^{l_1} \sum_{k_3=-l_2}^{l_2} \sum_{k_4=-l_2}^{l_2} \langle l_3\, k_5 \mid l_1\, k_1,\ l_2\, k_3 \rangle \langle l_3\, k_6 \mid l_1\, k_2,\ l_2\, k_4 \rangle [f]^{l_1}_{k_1 k_2}\, [f]^{l_2}_{k_3 k_4}. \tag{H5}$$

Under a rotation $\mathcal{R} \in \mathrm{SO}(3)$, the Wigner coefficients transform as

$$[f]^l_{k_1 k_2} \longrightarrow [f_{\text{rot}}]^l_{k_1 k_2} = \sum_{m=-l}^{l} D^l_{m k_1}(\mathcal{R})\, [f]^l_{m k_2}. \tag{H6}$$

Substituting these rotated coefficients into (H5) and using the properties of the Clebsch–Gordan coefficients (which couple the representations) along with the unitarity of the $D^l$ matrices, one obtains

$$[f^2_{\text{rot}}]^{l_3}_{k'_5 k_6} = \sum_{m=-l_3}^{l_3} D^{l_3}_{m k'_5}(g)\, [f^2]^{l_3}_{m k_6}. \tag{H7}$$

In other words, the squared function transforms as

$$f^2(\vec{r}) \longrightarrow f^2(\mathcal{R}^{-1}\vec{r}), \tag{H8}$$

which is exactly the equivariance property. Therefore, the entire local activation operator,

$$f_{\text{act}}(\vec{r}) = c_0 D + c_1 f(\vec{r}) + \frac{c_2}{D} f^2(\vec{r}), \tag{H9}$$

satisfies

$$f_{\text{act}}(\vec{r}) \longrightarrow f_{\text{act}}(\mathcal{R}^{-1}\vec{r}), \tag{H10}$$

proving its equivariance.

$$\square$$

## I DERIVATION OF EXPRESSIONS FOR ADAPTIVE COEFFICIENTS

The solution of Eq. 14 satisfies the following conditions,

$$\begin{cases} \partial F/\partial c_0 = 4c_0 + 4c_1 k + 4c_2(k^2 + \frac{1}{3}) - (k+1)^2 = 0 \\ \partial F/\partial c_1 = 4c_0 k + 4c_1(k^2 + \frac{1}{3}) + 4c_2(k^3 + k) - \frac{2}{3}(k+1)^3 = 0 \\ \partial F/\partial c_2 = 4c_0(k^2 + \frac{1}{3}) + 4c_1(k^3 + k) + 4c_2(k^4 + 2k^2 + \frac{1}{5}) - \frac{1}{2}(k+1)^4 = 0 \end{cases} . \tag{I1}$$

Multiplying the first equation by $k$ and $(k^2 + 1/3)$ and subtracting it from the second and third equations, respectively, we get

$$\begin{cases} c_1 + 2c_2 k = -\frac{1}{4}(k^3 - 3k - 2) \\ c_1 k + 2c_2(k^2 + \frac{1}{15}) = -\frac{1}{16}(3k^4 - 10k^2 - 8k + 1) \end{cases} . \tag{I2}$$

Figure I1 shows the plot of these coefficients as a function of normalized mean $k$ and the error of this approximation, respectively.

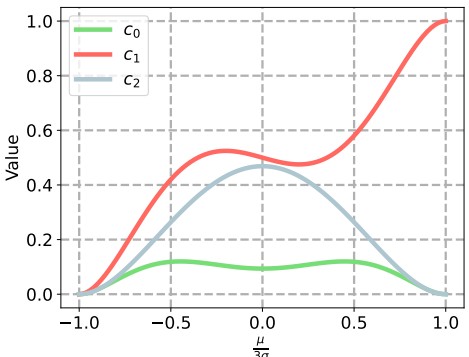 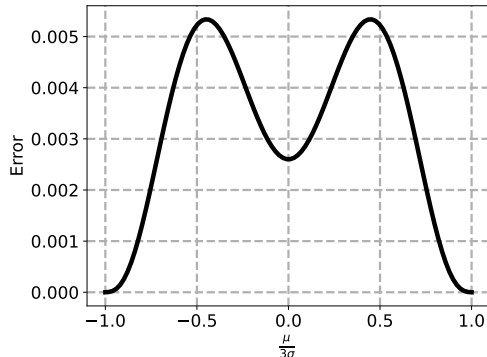

Figure I1: Left: Coefficients of the polynomial as functions of $\frac{\mu}{3\sigma}$. Right: Approximation error as a function of $\frac{\mu}{3\sigma}$.

## J    OTHER NON-LINEAL OPERATIONS

### J.1    MAX POOLING OPERATION IN THE CONTINUOUS SO(3) SPACE

For a wide range of tasks, whether they involve global or local (voxel-wise) prediction, it is necessary to design an operation that reduces the representation of the function in the rotation space to a single value. For brevity, we may call it a *pooling* operation in SO(3). The most straightforward approach, average pooling of a function in the SO(3) space, entails discarding Wigner coefficients of all degrees higher than zero. However, we aim to design an operation analogous to max pooling.

Let us note that the polynomial approximation from Eq. H1 allows for the simulation of the SoftMax effect in SO(3). More precisely, we can define SoftMax as

$$\text{SoftMax}(f) = \int_{\text{SO}(3)} w(\mathcal{R})\text{ReLU}(f(\mathcal{R}))d\mathcal{R}, \tag{J1}$$

where weight $w(\mathcal{R}) = \text{ReLU}(f(\mathcal{R}))/\int_{\text{SO}(3)} \text{ReLU}(f(\mathcal{R}'))d\mathcal{R}'$. The weight function estimates the ratio of $\text{ReLU}(f(\mathcal{R}))$ to the positive part of the function $f(\mathcal{R})$. Thus, we can also define the SoftMax function in SO(3) using only the Wigner matrix expansion coefficients of $f(\mathcal{R})$ as

$$\text{SoftMax}_{\text{poly}}(f) = \frac{\int_{\text{SO}(3)} \text{act}(f(\mathcal{R}))^2 d\mathcal{R}}{\int_{\text{SO}(3)} \text{act}(f(\mathcal{R}))d\mathcal{R}} = \frac{\|f_a\|_2^2}{\|f_a\|_1} = \frac{\sum_{l',k_1',k_2'} \frac{8\pi^2}{2l'+1}([f_a]_{k_1k_2}^l)^2}{[f_a]_{00}^0}. \tag{J2}$$

We should note that there can be different approaches to SoftMax simulation, even with the usage of polynomial coefficients. In scenarios where the pooling operation follows the ReLU operation, there is no need to apply the activation function again. Instead, we can directly calculate the ratio of the 2-norm squared of the input function to the pooling operation to its 1-norm.

### J.2    GLOBAL ACTIVATION

Above, we introduced a local activation function that requires a higher resolution and, consequently, an increased number of coefficients. This raises the question of whether the performance improvement offered by this operation justifies the associated rise in the computational cost. As a baseline for the activation, we explored an operation where the output resolution is identical to that of the input. The formalism presented above allows us to define the global activation,

$$f_{\text{GlobAct}}(\mathcal{R}) = \text{GlobAct}(f(\mathcal{R})) = \sigma(W\,\text{SoftMax}(f) + b)f(\mathcal{R}), \tag{J3}$$

where we use SoftMax from Eq. J2, apply $\sigma = $ sigmoid, and then multiply the result by the function $f(\mathcal{R})$. $W$ and $b$ are trainable parameters. Linearity of the Wigner matrix decomposition gives us the following expression for the Wigner coefficients of the global activation function,

$$[f_{\text{GlobAct}}]_{k_1k_2}^l = \sigma(W\,\text{SoftMax}(f) + b)[f]_{k_1k_2}^l. \tag{J4}$$

This expression can be seen as a gating mechanism that only passes SO(3)-distributions with sufficiently high trainable positive values, at least for a single point in SO(3). Although the input and output resolutions of this operation are identical, we still employ a higher resolution (expansion order) in the ReLU approximation within the SoftMax operation.

## J.3 NORMALIZATION

The fact that the mean and the standard deviation of a function in the rotational space can be expressed in terms of Wigner coefficients allows us to introduce an expression for the normalization of a function in SO(3):

$$f_n(\mathcal{R}) = \gamma \frac{f(\mathcal{R}) - \mu}{\sigma} + \beta, \tag{J5}$$

where $\mu$ and $\sigma$ are the mean and the standard deviation of function $f$, respectively, and $\gamma$ and $\beta$ are trainable coefficients. We can also extend this operation for data in $\mathbb{R}^3 \times \text{SO}(3)$, where the Euclidean component is discretized and characterized by a set of functions $f_{ijk}(\mathcal{R})$, where $i$, $j$, and $k$ are voxel indices. The batch normalization is then defined as follows:

$$[f_{ijk}]_n(\mathcal{R}) = \gamma \frac{f_{ijk}(\mathcal{R}) - \mu_{ijk}}{\sigma_{ijk}} + \beta, \tag{J6}$$

where $\mu_{ijk}$ and $\sigma_{ijk}$ are the mean and the standard deviation of a function $f_{ijk}$, respectively, and $\mu = \frac{1}{N} \sum_{i,j,k} \mu_{ijk}$ and $\sigma^2 = \frac{1}{N} \sum_{i,j,k} \sigma_{ijk}^2$, with $N$ being the number of voxels.

## K DATASETS, ARCHITECTURES AND TECHNICAL DETAILS

Our method is centered on applications involving *regular volumetric data*. Extending this method to irregular data would necessitate significant modifications that are beyond the scope of this paper. Consequently, we evaluated our method using a collection of voxelized 3D image datasets.

### K.1 DATASETS

We assessed our designs on MedMNIST v2, a vast MNIST-like collection of standardized biomedical images (Yang et al., 2023). It is designed to support a variety of tasks, including binary and multi-class classification, and ordinal regression. The dataset encompasses six sets comprising a total of 9,998 3D images. All images are resized to $28 \times 28 \times 28$ voxels, each paired with its respective classification label. We used the train-validation-text split provided by the authors of the dataset (the proportion is $7 : 1 : 2$).

### K.2 BASELINE ARCHITECTURES

For our baselines, we employed the same model configurations that the creators of the dataset utilized for testing on MedMNIST3D datasets(Yang et al., 2023). These include various adaptations of ResNet (He et al., 2016), featuring 2.5D/3D/ACS (Yang et al., 2021) convolutional layers, alongside open-source AutoML solutions such as auto-sklearn (Feurer et al., 2019), and AutoKeras (Jin et al., 2019). We have also added results of the models that were tested on the collection: FPVT (Liu et al., 2022a), Moving Frame Net (Sangalli et al., 2023), Regular SE(3) convolution (Kuipers and Bekkers, 2023) and ILPOResNet50 ((Zhemchuzhnikov and Grudinin, 2024)). Table 1 lists all the tested architectures.

### K.3 TRAINED MODELS

We developed multiple architectures with various activation methods, each reflecting the layer sequence of ResNet-18 (He et al., 2016). The final architecture, featuring reduced-resolution activation, is inspired by ResNet-50 (He et al., 2016). Figure K1 schematically illustrates our EquiLoPo architecture along with its main components. These include the initial block, the repetitive building block, pooling operators in SO(3) and 3D spaces, and a linear transformation at the end. Table K3 details the basic block for SE(3) data in ResNet-18. The initial convolutional block, outlined in

Table K2, initiates the architecture. We also specifically adapted the Batch Normalization process for $6D$ ($3D \times \mathrm{SO}(3)$) data.

It is important to note that the Bottleneck block in ResNet-50 (the simplest building block of the architecture), which increases activation resolution, results in an eightfold increase in the maximum degree. This surge is attributed to triple consecutive activations without an intervening convolution, potentially escalating computational demands. Consequently, our exploration was limited to the ResNet-18 architecture, where the activation operation enhances output resolution. Figure K1 illustrates the EquiLoPO ResNet-18 architecture.

scheme -02.pdf

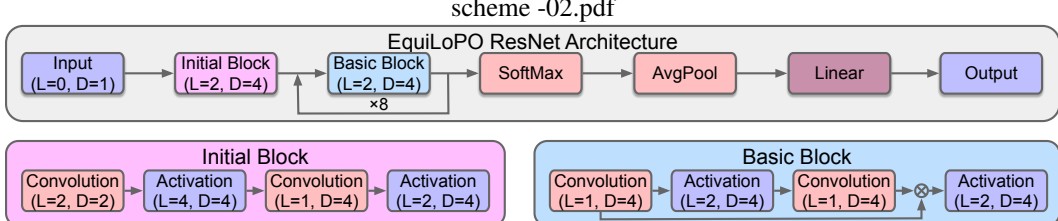

Figure K1: Schematic representation of the EquiLoPO ResNet-18 architecture, with a sequence of operations in the Initial and Basic blocks. $L$ is the maximum expansion degree of the last operator in the block and $D$ is the number of the block's features.

Every filter in our trained networks is confined within a $3 \times 3 \times 3$ volume in $3D$. Filters are parameterized by weights $w^{l_2 l_4}_{k_3 k_7 k_8}(r_i)$, where $r_i \in 0, 1, \sqrt{2}, \sqrt{3}$ represents all possible radii from the center in the cubic space. We used the ADAM optimizer (Kingma and Ba, 2014). In order to avoid overfitting, we apply the dropout operation after each activation operation. The learning rate of the optimizer and the dropout rate are hyperparameters. All the models are trained for 100 epochs. Table K1 lists hyperparameters optimized for validation data performance. Table K4 presents memory and time consumption metrics for EquiLoPOResNet-18, ILPOResNet-18 and ResNet-18 models in the inference mode. The current implementation consumes significantly more memory than the vanilla ResNet architecture and requires longer execution time because the activation is coded in Python rather than C++. Transitioning the implementation to C++ could significantly reduce both memory and CPU footprints.

| Methods | Organ | Nodule | Fracture | Adrenal | Vessel | Synapse |
|---|---|---|---|---|---|---|
| Local trainable activation | lr = 0.01, dr = 0.01 | lr = 0.01, dr = 0.00 | lr = 0.005, dr = 0.01 | lr = 0.005, dr = 0.00 | lr = 0.01, dr = 0.01 | lr = 0.01, dr = 0.00 |
| Local adaptive activation | lr = 0.01, dr = 0.01 | lr = 0.0005, dr = 0.01 | lr = 0.0005, dr = 0.00 | lr = 0.0005, dr = 0.01 | lr = 0.005, dr = 0.01 | lr = 0.01, dr = 0.01 |
| Local constant activation | lr = 0.01, dr = 0.01 | lr = 0.005, dr = 0.01 | lr = 0.01, dr = 0.01 | lr = 0.0005, dr = 0.01 | lr = 0.005, dr = 0.01 | lr = 0.005, dr = 0.01 |
| Global trainable activation | lr = 0.005, dr = 0.00 | lr = 0.005, dr = 0.01 | lr = 0.005, dr = 0.00 | lr = 0.005, dr = 0.00 | lr = 0.005, dr = 0.00 | lr = 0.005, dr = 0.00 |
| Global adaptive activation | lr = 0.005, dr = 0.01 | lr = 0.01, dr = 0.00 | lr = 0.01, dr = 0.00 | lr = 0.01, dr = 0.01 | lr = 0.005, dr = 0.00 | lr = 0.005, dr = 0.01 |
| Global constant activation | lr = 0.005, dr = 0.01 | lr = 0.005, dr = 0.00 | lr = 0.01, dr = 0.00 | lr = 0.005, dr = 0.00 | lr = 0.01, dr = 0.00 | lr = 0.0005, dr = 0.00 |
| Local trainable activation, const. resolution .([1]) | lr = 0.01, dr = 0.01 | lr = 0.01, dr = 0.01 | lr = 0.01, dr = 0.01 | lr = 0.01, dr = 0.01 | lr = 0.01, dr = 0.01 | lr = 0.01, dr = 0.01 |

Table K1: Optimal hyperparameters for the trained networks: learning rate (lr) and dropout rate (dr) of the trained networks. ([1])Here, we implemented the ResNet-50 architecture. In the other designs, we used the ResNet-18 versions.

| Step | Operation | Details (size of the filter in $3D$; output, input and filter maximum degrees) |
|---|---|---|
| 1 | Convolution, Eq. 8 | 3x3x3, $L_{\text{in}} = 0, L_{\text{out}} = 2, L_{\text{filter}} = 2$ |
| 2 | Batch Normalization, Eq. J6 | $L_{\text{in}} = 2, L_{\text{out}} = 2$ |
| 3 | Local Activation, Eq. H1 | $L_{\text{in}} = 2, L_{\text{out}} = 4$ |
| 4 | Convolution, Eq. 8 | 3x3x3, $L_{\text{in}} = 4, L_{\text{out}} = 1, L_{\text{filter}} = 2$ |
| 5 | Batch Normalization, Eq. J6 | $L_{\text{in}} = 1, L_{\text{out}} = 1$ |
| 6 | Local Activation, Eq. H1 | $L_{\text{in}} = 1, L_{\text{out}} = 2$ |

Table K2: Sequence of operations in the Initial convolutional block of EquiLoPOResNet-18.

| Step | Operation | Details (size of the filter in $3D$; output, input and filter maximum degrees) |
|---|---|---|
| 1 | Convolution, Eq. 8 | 3x3x3, $L_{\text{in}} = 2, L_{\text{out}} = 1, L_{\text{filter}} = 2$ |
| 2 | Batch Normalization, Eq. J6 | $L_{\text{in}} = 1, L_{\text{out}} = 1$ |
| 3 | Local Activation, Eq.H1 | $L_{\text{in}} = 1, L_{\text{out}} = 2$ |
| 4 | Convolution, Eq. 8 | 3x3x3, $L_{\text{in}} = 2, L_{\text{out}} = 1, L_{\text{filter}} = 2$ |
| 5 | Batch Normalization, Eq. J6 | $L_{\text{in}} = 1, L_{\text{out}} = 1$ |
| 6 | Addition | Add input ($l = 0, 1$) to the output |
| 7 | Local Activation, Eq. H1 | $L_{\text{in}} = 1, L_{\text{out}} = 2$ |

Table K3: Sequence of operations in the Basic block of EquiLoPOResNet-18.

| Model | Memory, GB | GFLOPs | Inference time per batch of 32 samples, seconds |
|---|---|---|---|
| ResNet-18 | 2.47 | 35.52 | 0.03 |
| ILPONet-18 | 7.14 | 19.58 | 0.3 |
| EquiLoPO (local activation) | 27.02 | 68.44 | 2.49 |
| EquiLoPO (global activation) | 30.05 | 53.75 | 2.13 |

Table K4: Memory, FLOPs and Time consumption for EquiLoPOResNet-18, ILPOResNet-18 and ResNet-18 inference.

## L PERFORMANCE WITH FEWER BUILDING BLOCKS

In the main experiments, we evaluated the proposed neural network architecture using the standard ResNet-18 configuration with 8 building blocks. To further investigate the method's performance and scalability, we conducted additional experiments by reducing the number of building blocks to 1 and 2, respectively, for the three activation functions that demonstrated the best performance in the main experiments. These were local activation with trainable coefficients(Table L1), local activation with adaptive coefficients(Table L2), and global activation with trainable coefficients(Table L3).

As the number of building blocks decreases from 8 to 2 and then to 1, there is a general trend of degrading the performance across all datasets and activation functions. The extent of performance degradation varies across datasets. For example, the OrganMNIST3D dataset exhibits a more significant drop in accuracy (ACC) when reducing the number of blocks compared to other datasets like NoduleMNIST3D or VesselMNIST3D.

The three activation functions (local activation with trainable coefficients, local activation with adaptive coefficients, and global activation with trainable coefficients) show different levels of robustness to the reduction in building blocks. Local activation with adaptive coefficients (Table 5) maintains relatively high performance even with fewer blocks, compared to the other two activation functions.

| Methods | # of prms | Organ | | Nodule | | Fracture | | Adrenal | | Vessel | | Synapse | |
|---|---|---|---|---|---|---|---|---|---|---|---|---|---|
| | | AUC | ACC | AUC | ACC | AUC | ACC | AUC | ACC | AUC | ACC | AUC | ACC |
| 8 Blocks | 418k | 0.991 | 0.866 | 0.923 | 0.861 | 0.727 | 0.563 | 0.876 | 0.792 | 0.950 | 0.958 | 0.965 | 0.878 |
| 2 Blocks | 176k | 0.968 | 0.705 | 0.902 | 0.855 | 0.716 | 0.525 | 0.892 | 0.785 | 0.962 | 0.927 | 0.838 | 0.832 |
| 1 Block | 136k | 0.936 | 0.489 | 0.872 | 0.861 | 0.726 | 0.508 | 0.885 | 0.846 | 0.963 | 0.914 | 0.876 | 0.861 |

Table L1: Performance comparison for ResNet-like architecture with 1, 2 and 8 blocks with local activation and trainable coefficients.

| Methods | # of prms | Organ | | Nodule | | Fracture | | Adrenal | | Vessel | | Synapse | |
|---|---|---|---|---|---|---|---|---|---|---|---|---|---|
| | | AUC | ACC | AUC | ACC | AUC | ACC | AUC | ACC | AUC | ACC | AUC | ACC |
| 8 Blocks | 418k | 0.977 | 0.767 | 0.916 | 0.871 | 0.803 | 0.613 | 0.885 | 0.805 | 0.968 | 0.958 | 0.984 | 0.940 |
| 2 Blocks | 176k | 0.970 | 0.689 | 0.929 | 0.871 | 0.725 | 0.513 | 0.885 | 0.832 | 0.970 | 0.935 | 0.946 | 0.866 |
| 1 Block | 136k | 0.961 | 0.659 | 0.914 | 0.852 | 0.786 | 0.596 | 0.909 | 0.862 | 0.963 | 0.932 | 0.920 | 0.824 |

Table L2: Performance comparison for ResNet-like architecture with 1, 2 and 8 blocks with local activation and adaptive coefficients.

| Methods | # of prms | Organ | | Nodule | | Fracture | | Adrenal | | Vessel | | Synapse | |
|---|---|---|---|---|---|---|---|---|---|---|---|---|---|
| | | AUC | ACC | AUC | ACC | AUC | ACC | AUC | ACC | AUC | ACC | AUC | ACC |
| 8 Blocks | 113k | 0.960 | 0.654 | 0.904 | 0.881 | 0.751 | 0.600 | 0.828 | 0.768 | 0.958 | 0.945 | 0.848 | 0.807 |
| 2 Blocks | 43k | 0.960 | 0.643 | 0.894 | 0.858 | 0.746 | 0.533 | 0.887 | 0.829 | 0.929 | 0.916 | 0.816 | 0.810 |
| 1 Block | 32k | 0.939 | 0.546 | 0.894 | 0.852 | 0.722 | 0.538 | 0.875 | 0.842 | 0.923 | 0.911 | 0.841 | 0.813 |

Table L3: Performance comparison for ResNet-like architecture with 1, 2 and 8 blocks with global activation and trainable coefficients.

## M  ALTERNATIVE STRATEGY FOR THE GLOBAL ACTIVATION

In this section, we test an alternative strategy for global activation. Instead of using Eq. J3, we use

$$f_{\text{GlobAct}}(\mathcal{R}) = \text{GlobAct}(f(\mathcal{R})) = \sigma(W|f|_2 + b)f(\mathcal{R}), \tag{M1}$$

where we replace the SoftMax function with the 2-norm of the function. Table M1 compares the performance of the ResNet-18-like architecture with global activation using trainable coefficients and the two different strategies: SoftMax and 2-norm. The results show that using the SoftMax function consistently outperforms the 2-norm alternative across all datasets.

| Methods | # of prms | Organ | | Nodule | | Fracture | | Adrenal | | Vessel | | Synapse | |
|---|---|---|---|---|---|---|---|---|---|---|---|---|---|
| | | AUC | ACC | AUC | ACC | AUC | ACC | AUC | ACC | AUC | ACC | AUC | ACC |
| SoftMax | 113k | 0.960 | 0.654 | 0.904 | 0.881 | 0.751 | 0.600 | 0.828 | 0.768 | 0.958 | 0.945 | 0.848 | 0.807 |
| 2-norm | 113k | 0.733 | 0.220 | 0.584 | 0.794 | 0.602 | 0.383 | 0.711 | 0.768 | 0.609 | 0.550 | 0.539 | 0.730 |

Table M1: Performance comparison for the ResNet-18-like architecture with global activation using trainable coefficients and different aggregation functions.

