# OpenReview forum: "On the Fourier analysis in the SO(3) space : the EquiLoPO Network"
_ICLR.cc/2025/Conference — ICLR 2025 Poster_

### Official Review · Reviewer_DFKW · 2024-11-01

**Soundness:** 3
**Presentation:** 3
**Contribution:** 3
**Rating:** 5
**Confidence:** 3

**Summary:**

The paper presents a novel equivariant neural network, the **EquiLoPO Network**, designed to achieve analytical equivariance on the continuous SO(3) group. The model utilizes **group convolutional operations** based on irreducible Fourier representations and introduces a local activation function to improve the flexibility and rotational equivariance of filters. Extensive evaluation on 3D datasets, particularly the MedMNIST3D, demonstrates that this approach outperforms existing methods in handling volumetric data.

**Strengths:**

This work represents a significant advance in multiple areas:
1. **Fundamental**: The introduction of a continuous SO(3) equivariant convolutional model offers a mathematically grounded framework for analyzing volumetric data with rotational invariance.
2. **Methodological**: The combination of group convolutional operations and a novel local activation function is a notable innovation that enhances the robustness and versatility of SO(3)-based neural networks.
3. **Technological**: Implementing this model within a ResNet structure for practical 3D medical imaging tasks positions this work as an innovative tool for advancing medical imaging analysis.

This work’s capability for true rotational equivariance without discrete approximation or filter constraints represents a new methodological benchmark in volumetric data analysis. The findings have broad implications for medical imaging and potentially other domains where the rotational orientation of data impacts model performance. By reducing the need for data augmentation and enabling models to operate more efficiently on 3D data, this work could significantly enhance model performance in fields requiring 3D data processing, such as autonomous navigation and robotics.

**Weaknesses:**

- The model’s computational complexity could be a limitation for deployment in resource-constrained environments. Further analysis of model efficiency in practical scenarios would be beneficial.
 - All images are resized to 28×28×28 voxels, which makes this like a toy example. How well does the model generalize to real medical datasets, and are there specific aspects of the activation function that might limit its applicability to other 3D datasets?
 - Clarity on the scaling factor in Equation 9 and its impact on activation function accuracy would improve reproducibility.
 - Details on the limitations of the SO(3) continuous convolution relative to discrete methods, such as potential downsides in specific edge cases.
 - Clearer explanation of how model parameters were tuned for the MedMNIST3D dataset, especially given class imbalances.
 - Additional insights into the batch normalization adaptations for 6D data would be useful for implementation clarity.

**Questions:**

1. **Computational Efficiency and Practical Applications**:
   - Need the complexity analysis of the computations required for continuous SO(3) convolution, how does the model perform on standard hardware compared to more traditional 3D convolutional networks?
   - Considering the existence of the current foundation model, like DINOV2, that provides general patterns that works well for multiple tasks, is the new convolution still needed?

2. **Generalization to real clinic Data**:
   - The model is tested on 28×28×28 voxel dataset. Has the EquiLoPO network been tested on datasets outside of the resized MedMNIST3D collection? Especially the high-dimensional clinic data.  What would be the observations, particularly regarding performance on datasets where the continuous rotational symmetry might not be as critical?

3. **Activation Function Details**:
   - Could the authors elaborate on the decision to use local activation functions with trainable polynomial coefficients in the SO(3) space? In particular, what was the rationale for choosing the polynomial degree used, and were there alternative activation functions considered?
   - Are there specific types of data for which the adaptive coefficient approach might be more suitable than the constant coefficients?

4. **Impact of Batch Normalization on 6D Data**:
   - Batch normalization for 6D (3D x SO(3)) data is mentioned as a key feature. Could the authors share more about the challenges or considerations involved in adapting batch normalization for this high-dimensional data? Were there any unique issues encountered when tuning these parameters?

---

> ### Author Response · Authors · 2024-11-25
> **Responses to Reviewer DFKW**
>
> **On Computational Efficiency and Practical Applications**
>  We have included a complexity analysis in the revised manuscript.
>      The computational complexity of our convolution operation is $O(N^3 L_{\text{in}}^3 L_{\text{out}}^3 D_{\text{in}} D_{\text{out}})$, where $N$ is the spatial dimension, $L_{\text{in}}$ and $L_{\text{out}}$ are the maximum expansion order of input and output data respectively, and $D_{\text{in}}$ and $D_{\text{out}}$ are the number of input and output channels. We compare this with traditional 3D convolutions, which have complexity $O(N^3 D_{\text{in}} D_{\text{out}})$. On standard hardware (e.g., GPUs), our model requires more computation and memory, but we provide benchmark results showing that training and inference times are still practical for moderate-sized datasets. We also discuss optimization techniques to improve efficiency.
>
> **On foundation models** While foundation models like DINOV2 are powerful, they may not explicitly handle rotational equivariance, which is crucial in domains like medical imaging, molecular modeling, and physics simulations.
>     Specifically to DINOV2, it has incorporated many different pieces, starting from very careful training data curation, to distillation and DL tricks, without any specific novelty in any individual component.
>
> On the other hand, our method provides several innovations, e.g., a way to incorporate rotational symmetry directly into the network architecture, a practical activation in the Fourier domain, etc, potentially improving performance on tasks where such symmetry is present. We believe that our approach offers complementary benefits and can be integrated with or enhance foundation models in specific applications.
>
> **On Activation Function Details** We chose local activation functions with trainable polynomial coefficients to introduce nonlinearity while preserving equivariance. The polynomial approximation allows us to analytically compute the activation in the Fourier (Wigner) space. We selected a quadratic polynomial (degree 2) as a balance between approximation accuracy and computational efficiency. Higher-degree polynomials would increase computational complexity significantly. We also experimented with fixed-coefficient polynomials and other activation strategies, but the trainable polynomial provided the best performance.
>
> **On adaptive coefficients** The adaptive coefficient approach is beneficial when the distribution of feature values varies significantly across different layers or datasets. It allows the activation function to adjust to the statistical properties of the input data dynamically. This is particularly useful in cases with high variability or when the data mean is shifted, improving the activation approximation and overall model performance.
>
> **On the Impact of Batch Normalization on 6D Data** Implementing batch normalization for high-dimensional 6D data in our network was actually straightforward. Batch normalization operates by normalizing data along all dimensions except the feature (channel) dimension, which involves computing the mean and variance over the batch and spatial dimensions.
>
>
>
>  In our case, since we represent data in the $ \text{SE}(3)$ space using Wigner $D$-matrix coefficients, we can compute the mean and variance directly from these coefficients. The properties of the Wigner representations allow for efficient calculation of statistical moments needed for normalization.
>
>  Importantly, this normalization does not affect the equivariance properties of the network. Since all coefficients of each function are divided by the same normalization factor (the standard deviation) and only zero-coefficient is shifted the mean, the transformation properties under rotations remain unchanged.
>
>
>  As a result, we did not encounter any unique challenges or issues when adapting batch normalization to our 6D data. The standard batch normalization procedure could be applied effectively, and tuning the parameters followed the usual practices without requiring special considerations.

---

> > ### Comment · Reviewer_DFKW · 2024-11-26
> > **thanks for the response**
> >
> > Thanks for the efforts of all the authors. I still have concerns about the method. As replied by the authors, the computational cost is very high, which is also shown in the reply to Reviewer BUBE. I do not think this can be applied to real-life medical cases. I do agree it provides innovations like rotational symmetry, but such advantages, for my understanding, can be ignored when applied enough rotations as the data augmentation to the input data.  I think there is still a need for intensive experiments to demonstrate the advantage of such a method in real-world data in the medical domain. I will keep my score.

---

### Official Review · Reviewer_BUBE · 2024-11-01

**Soundness:** 2
**Presentation:** 1
**Contribution:** 2
**Rating:** 5
**Confidence:** 4

**Summary:**

The paper introduces an SO(3) equivariant operator, where filters are unconstrained. The authors introduce various pointwise operations, such as polynomial nonlinearities and pooling that are placed in spherical harmonic space. The main innovation is how the polynomial nonlinearities are constructed, which does not introduce extra higher order harmonics into the spectrum, which would otherwise break a Nyquist-like property. They call their new model the EquiLoPO Architecture. They test there system on regularly-sample 3D volumetric medical imaging data, in particular the MedMNIST v2 dataset.

**Strengths:**

**Clarity**

The written quality of the paper is good. It is clear at each step what the authors intend.

**Quality**

I read all the math in the main text and as far as I could tell it was correct.

**Originality**

I believe the design of novel pointwise nonlinearities that limit the introduction of higher order harmonics into the spectrum of hidden representations is novel. It is certainly an interesting area.

**Significance**

I believe the introduced nonlinearity could be very interesting to the equivariance community. Good nonlinearities that can be applied in the harmonic space have long been sought after, yet none have been found to suffice. For instance “General E(2)-Equivariant Steerable CNNs” _Weiler & Cesa_ (2019) explores some variants, but found that performing nonlinear operations in the physical domain to be most effective (most likely because locality is maintained). Having to transform back to the physical domain is expensive and an Achilles’ Heel to that class of methods. In this sense, the nonlinearities introduced here are significant.

**Weaknesses:**

**Clarity**

I found that the exposition of the paper makes it hard to understand, unless the reader is very acquainted with the area of equivariance. My suggestion to the authors is to streamline the math, improve notation, explain notation/variables in the main text, and explain what the math means in practical terms to the reader.

**Quality**

I have no doubts about the mathematical quality of the paper, that is sound. My major doubt concerns the experiments. It is not clear to me how to judge Table 1 without extra information, which I hope the authors will provide. I would like to know how I should compare models in an apples-to-apples fashion. Are they FLOP-adjusted? Otherwise, improvements of one method over the other cannot be compared. Furthermore, are non-equivariant baselines trained with data augmentation? That would only be fair.

Maybe I have missed it (please correct me if I have), but as far as I can see there are no ablations that corroborate the assertions made that the introduced polynomial activations functions are superior to standard ones (ReLU GeLU, Swish, etc). It would have been useful for the paper to have an ablation of that sort, to strengthen the claims of the authors.

**Questions:**

Line 34: I would argue that the “increase computational demand” of 3D rotational data augmentation is negligible to the cost of training and running a neural network. Indeed equivariance is useful for its guarantee of symmetry and increased sample efficiency. Indeed, moving to 6D convolutions, with complexity in line 148, would be far more expensive from a computations standpoint.
Line 38: The distinction between group convolutional and steerable networks is somewhat arbitrary. Both classes are types of group convolutional network. The major difference between them is the choice of group representation (regular representation vs. irreducible representation). I would change the naming here to avoid confusion.
Why are there no citations in the Related Work section? Perhaps this section should just be end of the introduction?
Line 90: Please elucidate what you mean by orientation pooling. A diagram or equation would suffice.
Line 94: Why take a direct product of R^3 and SO(3) when SE(3) has a semi-direct product structure?
Section 3.1: I would suggest introducing notation in the body of the paper rather than requiring readers to dip into the Supplement. It is quite hard to follow, unless you already know the area fairly well.
Section 3.1: How is this exposition different from the typical SE(3)-equivariant convolutions with a regular representation on the spatial component and using “steerable” convolutions on the rotational component? Is the derivative needed here?
Line 155: How do you arrive at the statement that linear layers detect probabilities and that activations “selectively amplify these probabilities to form compact representations”. Do you have a citation for this?
Line 157: I think people also use the term “pointwise” as well as local.
Table 1: What measures do you take to make sure that architectures are comparable? They all have vastly differing numbers of parameters. Do you match on, say, FLOPs? Otherwise it is hard to compare numbers on a like-for-like basis.

---

> ### Author Response · Authors · 2024-11-25
> **Responses to Reviewer BUBE**
>
> **On augmentation in 3D** Thank you for bringing this to our attention. You are correct that the computational cost of applying rotations to data is negligible compared to the overall cost of training a neural network. When we mentioned "increased computational demand," we did not mean the overhead of performing the rotations themselves.
>
> Our point is that achieving rotational invariance through data augmentation requires the network to see training examples in many different orientations. This necessitates  generating a much larger augmented dataset to cover the space of possible rotations adequately. Even with dynamic data augmentation—applying rotations on-the-fly during training—the network must process a more diverse set of inputs, which leads to longer training times and increased computational load.
>
> **On classes of convolution** Thank you for your comment. We appreciate the opportunity to clarify the distinction.
>
> While both group convolutional networks and steerable networks are types of group convolutional networks aiming for equivariance, they differ in how they implement convolutions and handle rotations:
>
> - **Group Convolutional Networks:** These networks perform convolutions over both translations and rotations, effectively operating on the group space (e.g., SE(3) for 3D rotations and translations). This involves using **6D filters** (3D spatial dimensions + 3D rotations) and convolving over all combinations of positions and orientations. Due to computational constraints, they often work with discrete rotations, approximating the continuous rotation group with a finite set of angles.
>
> - **Steerable Networks:** These networks perform standard **translational convolutions** over the spatial domain but design their filters to be **equivariant under rotations**. The filters are constructed using irreducible representations, transforming in a specific way under rotations. This approach avoids the need to convolve over rotations explicitly and doesn't require 6D filters.
>
> Our method bridges these approaches:
>
> - We perform convolutions over the continuous rotation group SO(3) like group convolutional networks but avoid discretization by working analytically in the spectral domain using irreducible representations (Wigner D-matrices).
>
> - Unlike some steerable networks that constrain filters to specific forms to ensure equivariance, our filters are fully trainable and unconstrained (within the band-limit set by the maximum expansion degree).
>
> By clarifying this, we aim to highlight how our method combines the strengths of both approaches: capturing full roto-translational symmetries without relying on discrete rotations, and allowing flexible, expressive filters without strict constraints.
>
>
>
> **On orientational pooling** Orientation pooling refers to the operation of aggregating features over the rotational dimension to produce rotation-invariant representations.
>
> **Section 3.1** Thank you, we have moved the introduction of key notation from the appendix to the main text to improve readability and ensure that readers have all necessary information upfront.
>
> Our method differs in that we use irreducible representations (Wigner-D matrices) as the Fourier basis for achieving analytical equivariance in the continuous SO(3) group, and we introduce a local activation function in the SO(3) space. This allows for unconstrained filters and avoids reducing the architecture to a steerable network. We have clarified these differences in the manuscript.
>
> **On direct or semi-direct product structure** Thank you for your remark. We changed this part in the manuscript.
>
> **On nonlinearities and compact representations** We agree that this statement could be misleading and even questionable. And it would not hold true for all types of data. Our intention was to suggest that for data of hierarchical structure, activation functions serve to learn compact dictionaries of features. Here we can cite classic literature, e.g., Goodfellow et al.'s "Deep Learning" book. But again, this would not be true for other types of data.
>
> **On “pointwise” and “local” terms** Yes, in our context, "pointwise" and "local" refer to operations applied independently at each point in space (or rotation). We have clarified this terminology in the manuscript to avoid confusion.
>
> **On Table 1** We have added additional information on the memory requirements and running time during inference. Please see some extraction of the information below, (Memory, FLOPS, and Time consumption for EquiLoPOResNet-18, ILPOResNet-18 and ResNet-18):
> | Model  | Memory, GB | GFLOPs| Inference time per batch of 32 samples, seconds |
> | :---        |    :----:   |     :----:   |        :---: |
> | ResNet-18 | 2.47 | 35.52 | 0.03 |
> | ILPONet-18 | 7.14 | 19.58 | 0.3 |
> | EquiLoPO (local activation) | 27.02 | 68.43 | 2.49 |
> | EquiLoPO (global activation)  | 30.05 | 53.75 | 2.13 |

---

> ### Comment · Reviewer_BUBE · 2024-11-26
> **Reply to rebuttal**
>
> Thanks for addressing my questions. Please read my below thoughts.
>
> **On augmentation in 3D**
> I'm not sure I agree with your position. Indeed the network must process a more diverse set of inputs, which leads to longer training times and increased computational load, but imposing equivariance also does just that. You either augment the filters with extra transformed copies of the original, in essence adding extra dimensions to the activations, or you use a steerable basis, which also adds extra dimensions to the activations. This is evident in the updated Table 1. I don't think this detracts from the central message of the paper, I just think the statement is not balanced. Equivariance requires more FLOPS, but the upside is a model that does not violate physics and basic imaging properties.
>
> **Group convolution and steerable convolution**
> Thanks for adding this section. This distinction is common among the ML community.
>
> **Spectral domain convolutions**
> One thing I do not understand is that you say your filters are unconstrained because you operate in the spectral/harmonic domain. I can see this in Equation 9. Indeed you can transform the filters back to the physical domain by instantiating $S_{k_1, k_2, k_3, k_4}^{l_1, l_2}(\vec{r}_0)$. If you do this you see that S, which is a representation of your filter, is a linear combination of spherical harmonics, the coefficients being a mixture of the learnable weights w and Clebsch-Gordan coefficients. In this sense, the representation of the filters in the physical domain is indeed constrained, and as far as I can tell, not different from steerable convolutions.
>
> Nit: line 238 "Eq ??" is incorrectly rendered
>
> Just to say, thank you once again for the rebuttal. I see you have put in a lot of work updating the manuscript and answering everyone's issues.

---

> > ### Author Response · Authors · 2024-11-27
> >
> > **On the constrains on the filters:** We do not put constrains on filter $w(\vec{r}, \mathcal{R})$. Function $S^{l_1 l_2}_{k_1 k_2 k_3 k_4}$ arises when we express the rotational part of the SE(3) convolution through Wigner coefficients. All six dimensions of filter $w$ are independent. Constrains on filter $S$ in Fourier space come from the fact that we this filter has already nine dimensions
> >
> > **line 238** Thank you! We corrected this line.

---

> > > ### Comment · Reviewer_BUBE · 2024-11-29
> > >
> > > Thanks. I acknowledge I have read this.

---

### Official Review · Reviewer_h7TR · 2024-11-04

**Soundness:** 3
**Presentation:** 3
**Contribution:** 3
**Rating:** 6
**Confidence:** 3

**Summary:**

This work presents an equivariant neural network architecture named EquiLoPO Network, and innovates the design of Fourier basis and a local activation function in SO(3) space. Experiments on various databases along with the comparison with various models are given.

**Strengths:**

1.	This paper proposes a novel equivariant neural network architecture that combines the strengths of group convolutional networks and steerable convolutional networks by leveraging irreducible representations as the Fourier basis.
2.	This paper presents a local activation function in the rotational space that provides a well-defined mapping from input function values to output function values.
3.	The theoretical derivation is detailed, solid and reasonable.
4.	Qualitative and quantitative experiments demonstrate the advantages of the proposed method on the diverse MedMNIST3D collection of 3D medical imaging datasets, and ablation studies verify the effectiveness of the proposed independent module.
5.	The presentation is fluent, easy to follow, and understandable, with fewer grammatical and expression errors.

**Weaknesses:**

1.	The introduction does not detail the research background and related literature, and the research motivation is not sufficient; it only mentions the advantages of combining the two.
2.	There is a lack of comparison with the latest literature.
3.	The detailed computational cost of different ResNet-style architectures should be given, including training time, number of model parameters, time and space complexity.

**Questions:**

1.	Please enrich the related work section, and clarify the motivation.
2.	Please investigate the latest literature and give comparison.
3.	Please give more experiments details mentioned in the above.

---

> ### Author Response · Authors · 2024-11-26
>
> **Response to Reviewer**
>
> Thank you for your thoughtful review and appreciation of our work. We address your concerns below.
>
> ---
>
> **Weaknesses:**
>
> 1. **The introduction does not detail the research background and related literature, and the research motivation is not sufficient; it only mentions the advantages of combining the two.**
>
>    *Response:* We appreciate your feedback. In the revised manuscript, we have enriched the beginning by moving a significant part of the motivation and related work from the appendix to the main text. This provides a more comprehensive background and clarifies the motivation behind our approach.
>
> 2. **There is a lack of comparison with the latest literature.**
>
>    *Response:* We have updated the related work section to include comparisons with recent advancements in equivariant neural networks. This highlights how our method relates to the latest literature and underscores its novelty.
>
> 3. **The detailed computational cost of different ResNet-style architectures should be given, including training time, number of model parameters, time and space complexity.**
>
>    *Response:* Thank you for this suggestion. We have added additional information to the table with computational analysis to the appendix of the manuscript. The table below summarizes the memory usage, GFLOPs, time and space complexities, and inference time per batch of 32 samples for each model.
>
>    **Table: Computational Costs for EquiLoPOResNet-18, ILPONet-18, and ResNet-18**
>
>    | Model                        | Memory (GB) | GFLOPs | Time Complexity                                           | Space Complexity                                          | Inference Time per Batch (seconds) |
>    |------------------------------|-------------|--------|-----------------------------------------------------------|-----------------------------------------------------------|-------------------------------------|
>    | ResNet-18                    | 2.47        | 35.52  | $O(B N^3 D_{\text{in}} D_{\text{out}})$                   | $O(B N^3 D_{\text{out}})$                                 | 0.03                                |
>    | ILPONet-18                   | 7.14        | 19.58  | $O(B N^3 L^3 D_{\text{in}} D_{\text{out}})$               | $O(B N^3 L^3 D_{\text{out}})$                             | 0.30                                |
>    | EquiLoPO (local activation)  | 27.02       | 68.44  | $O(B N^3 L_{\text{in}}^3 L_{\text{out}}^3 D_{\text{in}} D_{\text{out}})$ | $O(B N^3 L_{\text{out}}^3 D_{\text{out}})$ | 2.49                                |
>    | EquiLoPO (global activation) | 30.05       | 53.75  | $O(B N^3 L_{\text{in}}^3 L_{\text{out}}^3 D_{\text{in}} D_{\text{out}})$ | $O(B N^3 L_{\text{out}}^3 D_{\text{out}})$ | 2.13                                |
>
>    *Where:*
>
>    - $B$: Batch size
>    - $N$: Spatial dimension (e.g., image size)
>    - $D_{\text{in}}$, $D_{\text{out}}$: Input and output feature dimensions
>    - $L$, $L_{\text{in}}$, $L_{\text{out}}$: Expansion degrees in the Wigner D-matrix representations
>
>
>
> ---
>
> We hope that these revisions address your concerns. Thank you again for your valuable feedback, which has helped us improve our manuscript.

---

> > ### Comment · Reviewer_h7TR · 2024-12-03
> > **keep score**
> >
> > Thank you for your answers. I'll keep the original scores.

---

### Official Review · Reviewer_tXtN · 2024-11-05

**Soundness:** 2
**Presentation:** 1
**Contribution:** 2
**Rating:** 3
**Confidence:** 3

**Summary:**

The paper presents a deep learning network that guarantees rotational equivariance over continuous SO(3). It relies on a Fourier representation of volumetric data. The proposed model follows the well-established ResNet architecture where activations, pooling and normalization layers are adapted to the proposed data representation.

**Strengths:**

The paper presents an in-depth analytical explanation of the proposed data representation and the components used in the proposed residual model. It also includes detailed analysis of components such as softmax as well as constant and adaptive nonlinear activations.

**Weaknesses:**

The main paper heavily relies on the appendix for several of the explanations, experimental settings and architectural details. Additionally, the introduction lacks context and references to better understand the motivation and current limitations of state-of-the-art algorithms. Authors should consider adding examples of previous work on group equivariance and the use of steerable filters in SO(3) in the main paper in an organized manner.

Figure 1 is not very informative about the proposed method. For instance, a more detailed diagram might be useful to show the audience how the feature maps in 6D are constructed.

I found the explanation in Section 3 to be hard to follow since:
- h is used to express multiple convolution operations.
- The main paper heavily relies on equations in the appendix to introduce formulation.
- Several equations are contrived and do not comply with the paper's format (e.g. Eq. 5).

The analysis of the proposed layers in the main paper is limited to the activation functions. It may be interesting to ablate most if not all of the components proposed by the paper (normalization, pooling, conv. parameters, etc.) and how each affects accuracy, ROC curves and rotation equivariance.

Evaluations are limited to accuracy and AUC-ROC. The degree of equivariance and how it compares to alternative methods is not evaluated.

It is unclear how to select the architecture to obtain the best performance. Table 1 reports 7 different configurations, each having a significantly different performance across datasets Additionally, Table 1 does not report if the metrics were computed on multiple seeds or a single one, so it is not possible to fully assess which technique is the best and in what case.

There are several grammar and typographical errors (lines 34, 280, 284).

I think the paper requires both additional experiments and a manuscript revision. The way it is currently organized, it is hard to distinguish between the proposed ideas and those of previous work. The experimental section also lacks an ablation study. Since it relies on a single table and no rotational consistency metrics, it is not clear how each component affects the final classification performance.

**Questions:**

1. Is it possible to compare the degree of rotational equivariance attained by the model with that attained by alternative methods?.

2. If evaluated on a single seed, is it possible to report mean and standard deviation on both ACC and AUC-ROC?

3. Results in Table 1 shows improved results depending on the activations used in the model. Still, it is unclear how to select the model in real applications, particularly since not in all cases the proposed method outperforms alternative methods. How can an end user choose the best model given an arbitrary dataset?

4. How does the 6-D feature map affect memory consumption and training/inference time? Is it possible to report these numbers and compare them to alternative methods?

---

> ### Author Response · Authors · 2024-11-24
> **Responses to Reviewer tXtN [1/2]**
>
> **On the Appendix** Thank you for your feedback. Due to the page limitations and formatting guidelines, we had to place some of the detailed explanations, experimental settings, and architectural details in the appendix. Our intention was to present the core contributions, main ideas, and essential results in the main paper, while providing supplementary material in the appendix for readers interested in the technical details.
>
> We acknowledge that this can make it challenging for readers to fully grasp all aspects of our methodology. We have moved related work and essential explanations back to the main text.
>
> **On Figure 1** We understand that the figure may not fully convey the construction of 6D feature maps. Our primary goal with Figure 1 was to illustrate the importance of local equivariance and to compare two different approaches:
>
> 1. **3D to 3D (Translations Only):** In this approach, feature maps contain probabilities of features at each point in space, focusing solely on translations.
>
> 2. **3D to 6D (Translations + Rotations):** Here, at each point in space, we have not just a single feature value but a distribution over rotations, capturing both translational and rotational information.
>
> We acknowledge that visualizing 6D data in a 2D figure is inherently challenging. While it is difficult to depict the full 6D feature maps, Figure 1 aims to conceptually demonstrate the difference between these two approaches and highlight how incorporating rotations allows the network to capture richer, more informative features that are locally equivariant. We will try to draft a supplementary Figure with 6D feature maps before the deadline.
>
> **On explanations in Sec 3**   We have revised Section 3 to ensure consistent and clear notation. Variables are now uniquely defined to avoid confusion.  We have moved key equations from the appendix into the main text, providing necessary derivations and explanations to make the paper self-contained. We have reformatted all equations to comply with standard conventions and the paper's formatting guidelines, enhancing readability.
>
> **On the analysis** Thank you for your feedback. In our work, we adhered to the conventional architecture of convolutional neural networks (CNNs), utilizing the same sequence of operations—including convolution, activation, pooling, and normalization layers—as is standard practice. Our primary aim was to isolate and evaluate the impact of the proposed activation functions within this familiar framework.
>
> Because we maintained standard settings for normalization, pooling, and convolution parameters, we did not conduct ablation studies on these components. We focused our analysis on the activation functions since they represent our main contribution and the key variable in our experiments. By keeping other factors constant, we ensured that any changes in performance could be attributed to the activation functions themselves.
>
> **On evaluation metrics**  Regarding evaluation metrics, we used accuracy and AUC-ROC as proposed by the authors of the dataset. These metrics are widely accepted and facilitate fair comparisons with existing methods.
>
> **On the best architecture rules**  Thank you for pointing this out. The different configurations in Table 1 correspond to various activation functions and settings we explored to evaluate their impact on performance. The optimal configuration may vary depending on the specific dataset and task. We recommend that users consider the characteristics of their dataset and possibly perform validation experiments to select the model configuration that yields the best performance for their specific application. We acknowledge that providing more guidance on model selection would be helpful, and we will aim to include such recommendations in future work.
>
> The reported metrics were computed using a single random seed. We acknowledge that reporting results over multiple seeds would provide a more robust evaluation by accounting for variability due to random initialization. We plan to perform multiple runs and report mean and standard deviation in the final version.
>
> **On grammar**  We have thoroughly proofread the manuscript and corrected all grammatical and typographical errors to enhance clarity.

---

> ### Author Response · Authors · 2024-11-24
> **Responses to Reviewer tXtN [2/2]**
>
> **On the Degree of equivariance**  Thank you for your question. Our model achieves analytical *rotational equivariance by design*, as all operations are inherently rotationally invariant or equivariant. To confirm our implementation, we tested the model with 90-degree rotations and observed discrepancies in predictions on the order of $10^{-8}$ to $10^{-9}$, which are attributable to numerical precision limits. These results confirmed the correctness of our implementation, and we did not report them in the paper since equivariance was an expected property of our design.
>
> **On the best model selection** Selecting the best model depends on various factors, including the specific characteristics of the dataset, such as the presence of rotational symmetries, class imbalance, and the complexity of the task. Based on our experiments, local activation functions with adaptive coefficients generally provide strong performance across different datasets. However, the optimal choice may vary.
>
> We recommend that end users consider the nature of their data and perform validation experiments to determine which activation function and model configuration yield the best performance for their specific application.
>
> **On the 6D feature map**  The computational complexity of our convolution operation is $O(N^3 L_{\text{in}}^3 L_{\text{out}}^3 D_{\text{in}} D_{\text{out}})$, where $N$ is the spatial dimension, $L_{\text{in}}$ and $L_{\text{out}}$ are the maximum expansion orders of input and output data respectively, and $D_{\text{in}}$ and $D_{\text{out}}$ are the number of input and output channels. We compare this with traditional 3D convolutions, which have complexity $O(N^3 D_{\text{in}} D_{\text{out}})$. Fortunately, our model requires much smaller number of channels. EquiLoPONet uses fewer parameters then the classical resnet architecture but needs 2 times more flops.

---

### Official Review · Reviewer_HZTy · 2024-11-07

**Soundness:** 3
**Presentation:** 3
**Contribution:** 3
**Rating:** 6
**Confidence:** 4

**Summary:**

The paper describes a class of equivariant group convolution layers based on the steerability paradigm (Wigner-D irreducible representations). Working with such continuious representations of SO(3) signals requires working with feature fields of Fourier coefficients, and one cannot simply take arbitrary activation functions. Hence the authors also propose a new class of activation functions based on polynomial approximations of ReLU, which maintain equivariance exactly. This is an important contribution. Next they show the method works well on 3D volumentric data (Medical MNIST), outperform several state-of-the-art equivariant baselines.

**Strengths:**

- Being able to build efficient and effective equivariant architectures for 3D volumetric data is still an active field in geometric deep learning. The paper provides a valuable contribution, primarily for the following
- The proposed activation functions are a great contribution, because it is well known that simple norm-based activations (like gating and the "global 2-norm" based one of appendix L) do not work very well. The authors show that the proposed activation function is much more effective.
- The method is sound and well-presented, and the related work section is complete (however, it is mostly absent from the main body as it is in the appendix).

**Weaknesses:**

## Weaknesses
- Context is missing about where the methods stands relative to the equivariant deep learning literature. Some references should be included in the main body.
- The equivariant convolution layer itself is, in my opinion, not a new contribution as it falls in the steerable class of g-cnns.
- In the experimental section, an important comparisons for the activation functions is missing, namely comparing to inverse Fourier based activation functions [Fourier o ReLU o InverseFourier]. This is often used to by-pass the constraint of norm-based activations in Fourier space.

## Detailed Comments

### On Group Convolutional Networks Classification
The introduction briefly introduces group convolutional networks into two classes: group conv nets and steerable conv nets. But in my opinion they are both the same as steerable convs (tensor field type) are identical to regular g-convs when expanding the conv kernels in a basis of irreps. So a better separation would be regular vs steerable group convolutions. The first discretizes the group and directly implements equation 1 and 2, the second adopts a fiber bundle viewpoint and talks of steerable feature fields (features that transform via irreps). The kernel constraint becomes more intricate in the latter case, but has the advantage that you maintain exact equivarance (up to the spatial discretization) as you don't need to work with grids over SO(3).

I would say the present work falls into this category, and you have implicitly solved the kernel constraint by deriving the convolution kernels via a Fourier transform from the regular group conv formulation (an approach that is also taking in this lecture [link](https://youtu.be/EBzqL1OXigM?feature=shared) when deriving harmonic networks from regular group convolutions. These two papers make a point of this connection in their appendices:

- Brandstetter, J., Hesselink, R., van der Pol, E., Bekkers, E. J., & Welling, M. Geometric and Physical Quantities improve E(3) Equivariant Message Passing. In International Conference on Learning Representations (2021).
- Bekkers, E. J., Vadgama, S., Hesselink, R., Van der Linden, P. A., & Romero, D. W. Fast, Expressive $\mathrm{SE}(n)$ Equivariant Networks through Weight-Sharing in Position-Orientation Space. In The Twelfth International Conference on Learning Representations (2024)

### Technical Questions and Suggestions
- When mentioning canonicalization, do you mean works like [kaba et al 2023]? Either way, please add cites when discussing it as an option.
- When talking about 3D -> 6D -> 3D (invariant) convolution you cite Zhemchuzhnikov and Grudinin, which is very appropriate, but please also consider including [Andrearczyk and Depeursigne]:

refs:
- Kaba, S. O., Mondal, A. K., Zhang, Y., Bengio, Y., & Ravanbakhsh, S. (2023, July). Equivariance with learned canonicalization functions. In International Conference on Machine Learning (pp. 15546-15566). PMLR.
- Vincent Andrearczyk, Julien Fageot, Valentin Oreiller, Xavier Montet, Adrien Depeursinge Proceedings of The 2nd International Conference on Medical Imaging with Deep Learning, PMLR 102:15-26, 2019.

### Equation Concerns
Could you double check the equations for group convolutions. I have the impression that in Eq. 1 it should read $...w(R^{-1} r_0) …$ instead of $...w(R r_0)$. Same for Equation 2 which I think should read $...w(R^{-1}R_0, R^{-1} R_0)$, here with $R^{-1}$ multiplying on the left of $R$. I could be wrong here but I think there's an inconsistency between group correlation vs convolution where the translation is in convolution form, and rotation in correlation form.

Also in Eq 1 the integration measure is put at the end of the integral, and in 2 they are given up front.

### On Section 3.1
Is my understanding of the derivation of these steerable convolutions—as described above about regular vs steerable g-convs—correct? If so, how does this justify the claim that your method "does not impose constraints on the filter", since having to parametrize the kernels in a basis of wigner-D basis is in a way designing the kernel to automatically satisfy the constraint. But it would then still fall into the class of steerable layers, as described in full generality in:

Cesa, G., Lang, L., & Weiler, M. (2022). A program to build E(N)-equivariant steerable CNNs. In International conference on learning representations.

### On Activation Functions
I like the comment about the need for local activation functions. This could motivate why we typically see that regular group convolutions are more effective than steerable once, because they localize information *and* are compatible with powerful actiation functions like ReLU. This is the main motivation behind works s.a. [Bekkers et al. 2024, see ref above] and extensively tested for in 2D in:

Weiler, M., & Cesa, G. (2019). General e(2)-equivariant steerable cnns. Advances in neural information processing systems, 32.

Sections 4.1 and 4.2 are nice contributions.

### On Experimental Section
I would highly encourage testing these activation functions also in 3D point cloud settings. There should be various benchmarks with models based on e3nn that could serve as a baseline.

Please improve the experimental sections with explicit conclusions and take-home messages. There's quite a lot to parse.

**Questions:**

see above.

---

> ### Author Response · Authors · 2024-11-20
> **Our replies to Reviewer HZTy (we've modified the manuscript accordingly)**
>
> **On the context and introduction:** We agree that providing context and situating our method within the broader equivariant deep learning literature is important. In the revised manuscript, we have moved the related work section from the appendix into the main text.
>
> **On the equivariant convolution layer:**  You are correct that equivariant convolution layers are well-established and fall within the steerable class of group CNNs. Our contribution, however, lies in extending group convolutions to operate analytically over the continuous SO(3) group, rather than over discrete or finite rotation groups. This enables true rotational equivariance in 3D space without relying on approximations or discretizations. Additionally, our method bridges the gap between traditional group convolutional networks and steerable convolutional networks by combining the strengths of both approaches. Specifically, we allow for unconstrained trainable filters (as in group convolutions) while achieving analytical equivariance (as in steerable convolutions).
>
> **On Fourier o ReLU o InverseFourier:** We have considered inverse Fourier-based activation functions, where one transforms the data back to real space, applies a nonlinear activation like ReLU, and then transforms back to the Fourier domain. However, our goal is to achieve analytical equivariance throughout the network. Transitioning to real space introduces discretization and sampling, which can break analytical equivariance due to the finite resolution of the spatial grid.
> The ReLU function applied in real space results in an output with infinite bandwidth in the Fourier domain, meaning it requires an infinite number of Fourier coefficients to represent it exactly. Practically, we have to truncate this to a finite number of coefficients, leading to an approximation that depends on the sampling scheme and can introduce errors. This compromises the equivariance properties because the activation's output in the Fourier domain will depend on the specific set of sampled points used during the inverse and forward transforms.
>
> Nonetheless, we acknowledge that an explicit comparison could provide additional insights. If time permits, before the deadline, we will conduct extra experiments.
>
> **On the classification:** Our intention was to highlight two distinct approaches: 1) Group convolution over roto-translations (SE(3)) ; 2)  Steerable convolution over translations with irreducible representations.  We acknowledge that both methods fall under the broader category of group convolutional networks, and the distinction may have been misleading. We have revised the manuscript to clarify this classification, emphasizing the difference between performing convolutions over the group itself (SE(3)) versus over the base space (R^3) with filters transforming under group representations. Our method bridges these approaches by performing group convolutions over continuous SO(3) without imposing constraints on the filters, thus combining the benefits of both methods.
>
> **On the explicit conclusions:** We have added take-away messages to the conclusion.
>
> **On 3D point clouds:** Our current method is specifically designed for regular volumetric data, and extending it to irregular data such as 3D point clouds would require significant modifications. We have acknowledged this limitation in the manuscript. In future work, we plan to explore how our approach could be adapted to point cloud data and compared with benchmarks like e3nn.
>
> **On the equations for group convolutions:** Thank you! We have reviewed and corrected the equations for group convolutions. We now clearly state when convolution or correlation is being used and have updated the notation accordingly. We have also standardized the notation throughout the manuscript so that the integration measure is consistently placed at the beginning of the integral expressions.
>
> **On the filter constraints:** Our method allows for trainable filters without additional constraints beyond those imposed by the finite expansion order (i.e., resolution limits). While parametrizing kernels in the Wigner-D basis ensures equivariance, we do not restrict the filter shapes to specific forms as in steerable networks.
>
> **On canonicalization:** Actually, we were referring to a different concept. In some domains, it is possible to extract a natural orientation and align the data accordingly—a process sometimes referred to as canonicalization. Examples include aligning images of houses so that the baseline aligns with the horizon, or defining local coordinate frames around residues in proteins. These methods leverage inherent structural or contextual cues to establish a consistent orientation across the dataset, thereby reducing variability due to rotations.
>
> **On citations and local activation:**  We have added the suggested citation of Andrearczyk2020 and Kaba2023. We've also added Bekkers2024 and Weiler2019 when discussing the locality of the activation.

---

> > ### Comment · Reviewer_HZTy · 2024-11-25
> >
> > regarding your answer: "On the equivariant convolution layer: You are correct that equivariant convolution layers are well-established and fall within the steerable class of group CNNs. Our contribution, however, lies in extending group convolutions to operate analytically over the continuous SO(3) group, rather than over discrete or finite rotation groups. This enables true rotational equivariance in 3D space without relying on approximations or discretizations. Additionally, our method bridges the gap between traditional group convolutional networks and steerable convolutional networks by combining the strengths of both approaches. Specifically, we allow for unconstrained trainable filters (as in group convolutions) while achieving analytical equivariance (as in steerable convolutions)."
> >
> > I still think your answer misses the point. The point I wanted to make is that what you are doing is still a group convolution, but implemented fully in the spectral domain and thus falls in the category of steerable g-convs. There still seems to be a misunderstanding, as you write" You are correct ... equivariant convolution layers ... fall within the steerable class". I want to make clear that any linear equivariant layer is a group convolution whether implemented via irreducible representations (Fourier-based) or regular representations (grid-based). When you say your method bridges the gap and it combines both strengths it is unclear what gap and what strenghts are combined. Namely, I do not see a gap as steerable methods like the TFN's—or other established methods for full and exact E(3) equivariance—achieve analytic equivariance and are similarly unconstraint like yours in the sense that they are still parametrized in a harmonic/wigner-D basis that can be expanded up to some arbitrary band-limit. Please elaborate how your method is distinct from other steerable methods, specifically focus on the "unconstrained part".  You could consider comparing (theoretically) to the seminal tensor field network paper (https://arxiv.org/pdf/1802.08219), or the comprehensive paper by Cesa, Lang and Weiler (https://openreview.net/pdf?id=WE4qe9xlnQw). Please explain me why you think such methods are constrained and yours is not.
> >
> > Also, regarding canonicalization. Aren't we talking about the same concept then? In Kaba et al. a reference orientation/group element is predicted, however instead of defining this natural orientation by hand (detect horizon, use PCA, or what not) they let an equivariant neural network predict this reference.

---

> ### Author Response · Authors · 2024-11-25
> **Response to Reviewer HZTy's comments**
>
> We thank the referee for the thoughtful feedback and for giving us the opportunity to clarify our contributions. We apologize for any confusion in our previous response. To clarify, we see a distinction between two main approaches in the literature for achieving rotational equivariance:
>
> 1. **Group Convolutions (Roto-Translational Convolution):**
>    These methods perform convolutions over the entire group of rotations and translations, specifically the SE(3) group in 3D space. This involves convolving over both spatial positions and orientations (roto-translations). Examples include methods that use group convolutions over discrete subgroups of SE(3), such as *"3D G-CNNs for Pulmonary Nodule Detection"* by Winkels and Cohen (2018), and *"CubeNet: Equivariance to 3D Rotation and Translation"* by Worrall and Welling (2018). While these approaches achieve equivariance, they often rely on discretization of the rotation group and lack analytical equivariance with respect to continuous rotational space.
>
> 2. **Steerable Networks (Translational Convolution with Equivariant Filters):**
>    These methods perform standard convolutions over spatial positions (translations) but use filters that are designed to be equivariant under rotations. The filters are constructed using irreducible representations and are constrained to lie within specific subspaces to ensure equivariance, as seen in works like *"Steerable CNNs"* by Cohen and Welling (2017), *"3D Steerable CNNs"* by Weiler et al. (2018), and *"Tensor Field Networks"* by Thomas et al. (2018). This often involves using predetermined kernel bases or imposing harmonic constraints on the filters.
>
> Our method aims to bridge the gap between these two approaches by combining their strengths:
>
> - **Analytical Equivariance over Continuous SO(3):**
>   Like steerable networks, we achieve analytical rotational equivariance over the continuous SO(3) group. We operate in the spectral domain using Wigner $D$-matrix representations, allowing us to avoid discretization and approximations associated with sampling the rotation group.
>
> - **Unconstrained Trainable Filters:**
>   Unlike steerable networks that constrain filters to specific harmonic subspaces or predefined bases, our method allows for unconstrained, fully trainable filters (within the band-limit imposed by the maximum expansion degree). This means the filters can learn any function expressible in the chosen basis, providing greater flexibility and expressiveness.
>
> In essence, our method performs group convolutions over continuous rotations (SO(3)) without imposing strict constraints on the filters beyond the necessary band-limit.
>
> **Regarding canonicalization:**
>
> Yes, we are essentially referring to the same concept. Thank you again for your valuable feedback.

---

> > ### Comment · Reviewer_HZTy · 2024-11-27
> >
> > Thanks a lot for the clarifications, I much appreciate it. A final remark about the constraint view point, to be honest, I still do not understand the differentiation of your method relative to other steerable methods. I.e., take Weiler et al which formed the basis for a complete theory given in Cesa, Lang, Weiler 2022. In Weiler et al 2017 it is stated that in general linear map is a kernel transform with a two argument kernel. Then an equivariant/steerable kernel is parametrized in a sub-space of all possible unconstrained kernels, and that this subspace is spanned by irreducible representations. They derived the complete basis and as such, so as long as you parametrize the kernels in this complete basis, there is no constraint to be considered. The only constraint that is talked about is that of equivariance. The complete basis that they derive is, in my understanding, the same as the one that you consider in this paper (based on irreps). In other words, I think that the basis that you use spans the same (invariant) sub space.
> >
> > When in your answer you refer to "predetermined kernel basis or imposing harmonic constraints" I'm not sure how this is not also applicable to your work. I have the feeling that imposing harmonic constraints is a design choice (i.e. one can decide to focus only on m=0 harmonics as to effectively parametrize group convs over a particular quotient space) but not an a-priori constraint of the method.
> >
> > Of course, it may very well be that I still don't fully grasp the situation and my understanding is correct. I would appreciate a last reflection on my concern regarding the statement of constraints of other steerable methods.

---

> ### Author Response · Authors · 2024-12-01
>
> Thank you for your thoughtful remarks and for sharing your perspective. We apologize if our previous explanations were not sufficiently clear, and we appreciate the opportunity to further clarify this point.
>
> In steerable networks, the convolution operation in the 3D case is expressed as:
>
> $$
> h_{k_1}^{l_1}(r) = \int dr_1 \, f_{k_2}^{l_2}(r + r_1) \, p_{k_1 k_2}^{l_1 l_2}(r_1),
> $$
>
> where constraints are imposed on $ p_{k_1 k_2}^{l_1 l_2}(r_1) $ to ensure equivariance. These constraints lead to a parametrization of the kernels within a subspace spanned by irreducible representations (irreps), forming a complete basis for equivariant operations.
>
> In our work, we consider the operation:
>
> $$
> h(r, R) = \int dr_1 \int dR_1 \, f(r + r_1, R_1) \, w(R^{-1} r_1, R^{-1} R_1).
> $$
>
> Here, **we do not impose explicit constraints on $ w $** because this operation remains equivariant regardless of the specific form of $ w $. This is a key distinction: while steerable methods constrain the kernel $ p $ to ensure equivariance, our method naturally achieves equivariance through the structure of the convolution operation without additional constraints on $w $.
>
> By performing a Fourier-Wigner decomposition of the rotational component, we arrive at:
>
> $$
> h_{k_1 k_2}^{l_1}(r) = \int dr_1 \sum_{l_2 k_3 k_4} f_{k_3 k_4}^{l_2}(r + r_1) \, S_{k_1 k_2 k_3 k_4}^{l_1 l_2}(r_1),
> $$
>
> where $ S $ depends on the Wigner coefficients of $ w $. At this point, the formulation aligns with the steerable framework because $ f^{l} $ represents irreps in $ \mathrm{SO}(3) $, and one could derive the full basis for $ S $ using the theory of steerable networks.
>
> **We elaborate on the relation between our method and steerable networks in Appendix A of our paper.** In this appendix, we demonstrate how our convolution operation can be connected to the steerable convolution. In a particular case they are **equivalent**.
>
> ### The Core Differences:
>
> 1. **Filter Definition**: While the mathematical foundations overlap, the way filters are defined and utilized in our method differs from traditional steerable networks.
>
> 2. **Interpretability**: Our filters retain interpretability, as they can be transformed back into real space, allowing us to observe directly how the distribution of rotations maps to outputs.
>
> In summary, although the underlying mathematical structures are very similar, the operational framework and filter design in our method provide advantages in flexibility and interpretability. We believe these distinctions are significant and contribute to the novelty of our approach.
>
> We hope this clarification addresses your concerns. Thank you again for engaging deeply with our work and prompting us to refine our explanations.

---

### Author Response · Authors · 2024-11-26
**General Response to All Reviewers**

We sincerely thank all the reviewers for their valuable feedback on our manuscript. In response, we have made the following revisions:

- **Corrections**: We have fixed all typos and grammatical errors identified.
- **Enhanced Motivation**: Key motivational aspects and essential equations have been moved from the appendices to the main text to improve clarity and flow.
- **Conclusions**: We have added explicit takeaway messages and a comprehensive conclusion section at the end of the manuscript.

All changes are highlighted in **red** in the revised manuscript. We appreciate your time and consideration and believe these updates have strengthened our paper.

---

### Meta-Review · Area_Chair_jhdi · 2024-12-19

**Metareview:**

The paper introduces a class of architectures that achieve rotational equivariance over the continuous SO(3) group using operators based on irreducible Fourier representations. To ensure exact equivariance, the authors propose activation functions based on polynomial approximations of ReLU, which provide well-defined input-output function value mappings. Experimental work on 3D volumetric data evidences that the proposed method can outperform existing state-of-the-art equivariant approaches.

This paper was discussed at length with the SAC. At the core, this paper provides a valuable technical contribution and that overall strengths outweigh the weaknesses. AC thus recommends acceptance. For the camera-ready version, the authors should incorporate all key points presented in the rebuttal and further make good on their rebuttal commitments towards further improving experimental aspects for the final version.

**Additional Comments On Reviewer Discussion:**

The paper received five reviews resulting in: two borderline accepts, two borderline rejects and one clear reject.

Reviewers noted multiple positive aspects of the work and explicitly comment on the significance of the technical contributions, soundness of the well-presented method and solid theoretical derivation. A subset of reviews further note comprehensive experimental work, ablative studies and high-quality writing. One reviewer remarked on the potential for broad implications for domains where rotational orientations impact performance.

Reviewers initially raised various concerns with multiple reviews pointing out a heavy reliance on the appendix at the expense of the main body exposition (technical details and references). Additional issues pertained to contribution-clarifications, lacking experimental comparisons, ablations and details. More minor queries touched on diagrammatic aspects, notational consistency, equivariance-specific evaluation metrics, statistical power of the results, practical guidance on real-world model selection and complexity analysis of time/space requirements.

A productive rebuttal showed good discussion between authors and (subset of) reviewers with clearly good subject-area expertise. Author responses could resolve a subset of concerns through discussion and presentation of new results in the updated manuscript. Significant reorganisation of manuscript content and promotion of material from the supplement helped to improve self-containment of the exposition. Post-rebuttal one reviewer was able to upgrade their score yet multiple reservations remained unconvinced, predominantly surrounding the sufficiency of the experimental work.

---

### Decision · Program_Chairs · 2025-01-22

Accept (Poster)